# Communication-Efficient Distributed Learning via Lazily Aggregated Quantized Gradients

**Jun Sun**[†]
Zhejiang University
Hangzhou, China 310027
sunjun16sj@gmail.com

**Tianyi Chen**[†]
Rensselaer Polytechnic Institute
Troy, New York 12180
chent18@rpi.edu

**Georgios B. Giannakis**
University of Minnesota, Twin Cities
Minneapolis, MN 55455
georgios@umn.edu

**Zaiyue Yang**
Southern U. of Science and Technology
Shenzhen, China 518055
yangzy3@sustc.edu.cn

## Abstract

The present paper develops a novel aggregated gradient approach for distributed machine learning that adaptively compresses the gradient communication. The key idea is to first *quantize* the computed gradients, and then *skip* less informative quantized gradient communications by reusing outdated gradients. Quantizing and skipping result in 'lazy' worker-server communications, which justifies the term **L**azily **A**ggregated **Q**uantized gradient that is henceforth abbreviated as **LAQ**. Our LAQ can provably attain the same linear convergence rate as the gradient descent in the strongly convex case, while effecting major savings in the communication overhead both in transmitted *bits* as well as in communication *rounds*. Empirically, experiments with real data corroborate a significant communication reduction compared to existing gradient- and stochastic gradient-based algorithms.

## 1 Introduction

Considering the massive amount of mobile devices, centralized machine learning via cloud computing incurs considerable communication overhead, and raises serious privacy concerns. Today, the widespread consensus is that besides in the cloud centers, future machine learning tasks have to be performed starting from the network edge, namely devices [17, 19]. Typically, distributed learning tasks can be formulated as an optimization problem of the form

$$\min_{\boldsymbol{\theta}} \ \sum_{m \in \mathcal{M}} f_m(\boldsymbol{\theta}) \ \text{ with } \ f_m(\boldsymbol{\theta}) := \sum_{n=1}^{N_m} \ell(\mathbf{x}_{m,n}; \boldsymbol{\theta}) \tag{1}$$

where $\boldsymbol{\theta} \in \mathbb{R}^p$ denotes the parameter to be learned, $\mathcal{M}$ with $|\mathcal{M}| = M$ denotes the set of servers, $\mathbf{x}_{m,n}$ represents the $n$-th data vector at worker $m$ (e.g., feature and label), and $N_m$ is the number of data samples at worker $m$. In (1), $\ell(\mathbf{x}; \boldsymbol{\theta})$ denotes the loss associated with $\boldsymbol{\theta}$ and $\mathbf{x}$, and $f_m(\boldsymbol{\theta})$ denotes the aggregated loss corresponding to $\boldsymbol{\theta}$ and all data at worker $m$. For the ease in exposition, we also define $f(\boldsymbol{\theta}) = \sum_{m \in \mathcal{M}} f_m(\boldsymbol{\theta})$ as the overall loss function.

In the commonly employed worker-server setup, the server collects local gradients from the workers and updates the parameter using a gradient descent (GD) iteration given by

**GD iteration** $$\boldsymbol{\theta}^{k+1} = \boldsymbol{\theta}^k - \alpha \sum_{m \in \mathcal{M}} \nabla f_m(\boldsymbol{\theta}^k) \tag{2}$$

---

[†] Jun Sun and Tianyi Chen contributed equally to this work.

where $\boldsymbol{\theta}^k$ denotes the parameter value at iteration $k$, $\alpha$ is the stepsize, and $\nabla f(\boldsymbol{\theta}^k) = \sum_{m \in \mathcal{M}} \nabla f_m(\boldsymbol{\theta}^k)$ is the aggregated gradient. When the data samples are distributed across workers, each worker computes the corresponding local gradient $\nabla f_m(\boldsymbol{\theta}^k)$, and uploads it to the server. Only when all the local gradients are collected, the server can obtain the full gradient and update the parameter. To implement (2) however, the server has to communicate with *all* workers to obtain fresh gradients $\{\nabla f_m(\boldsymbol{\theta}^k)\}_{m=1}^M$. In several settings though, communication is much slower than computation [16]. Thus, as the number of workers grows, worker-server communications become the bottleneck [10]. This becomes more challenging when incorporating popular deep learning-based learning models with high-dimensional parameters, and correspondingly large-scale gradients.

## 1.1 Prior art

Communication-efficient distributed learning methods have gained popularity recently [10, 22]. Most popular methods build on simple gradient updates, and are centered around the key idea of gradient compression to save communication, including gradient *quantization* and *sparsification*.

**Quantization.** Quantization aims to compress gradients by limiting the number of bits that represent floating point numbers during communication, and has been successfully applied to several engineering tasks employing wireless sensor networks [21]. In the context of distributed machine learning, a 1-bit binary quantization method has been developed in [5, 24]. Multi-bit quantization schemes have been studied in [2, 18], where an adjustable quantization level can endow additional flexibility to control the tradeoff between the per-iteration communication cost and the convergence rate. Other variants of quantized gradient schemes include error compensation [32], variance-reduced quantization [34], quantization to a ternary vector [31], and quantization of gradient difference [20].

**Sparsification.** Sparsification amounts to transmitting only gradient coordinates with large enough magnitudes exceeding a certain threshold [27]. Empirically, the desired accuracy can be attained even after dropping 99% of the gradients [1]. To avoid losing information, small gradient components are accumulated and then applied when they are large enough. The accumulated gradient offers variance reduction of the sparsified stochastic (S)GD iterates [12, 26]. With its impressive empirical performance granted, except recent efforts [3], deterministic sparsification schemes lack performance analysis guarantees. However, randomized counterparts that come with the so-termed unbiased sparsification have been developed to offer convergence guarantees [28, 30].

Quantization and sparsification have been also employed simultaneously [9, 13, 14]. Nevertheless, they both introduce noise to (S)GD updates, and thus deteriorate convergence in general. For problems with strongly convex losses, gradient compression algorithms either converge to the neighborhood of the optimal solution, or, they converge at sublinear rate. The exception is [18], where the first linear convergence rate has been established for the quantized gradient-based approaches. However, [18] only focuses on reducing the required bits per communication, but not the total number of rounds. Nevertheless, for exchanging messages, e.g., the $p$-dimensional $\boldsymbol{\theta}$ or its gradient, other latencies (initiating communication links, queueing, and propagating the message) are at least comparable to the message size-dependent transmission latency [23]. This motivates reducing the number of communication rounds, sometimes even more so than the bits per round.

Distinct from the aforementioned gradient compression schemes, communication-efficient schemes that aim to reduce the number of communication rounds have been developed by leveraging higher-order information [25, 36], periodic aggregation [19, 33, 35], and recently by adaptive aggregation [6, 7, 11, 29]; see also [4] for a lower bound on communication rounds. However, whether we can save communication bits and rounds simultaneously without sacrificing the desired convergence properties remains unresolved. This paper aims to address this issue.

## 1.2 Our contributions

Before introducing our approach, we revisit the canonical form of popular quantized (Q) GD methods [24]-[20] in the simple setup of (1) with one server and $M$ workers:

**QGD iteration** $$\boldsymbol{\theta}^{k+1} = \boldsymbol{\theta}^k - \alpha \sum_{m \in \mathcal{M}} Q_m(\boldsymbol{\theta}^k) \tag{3}$$

where $Q_m(\boldsymbol{\theta}^k)$ is the quantized gradient that coarsely approximates the local gradient $\nabla f_m(\boldsymbol{\theta}^k)$. While the exact quantization scheme is different across algorithms, transmitting $Q_m(\boldsymbol{\theta}^k)$ generally requires

fewer number of bits than transmitting $\nabla f_m(\boldsymbol{\theta}^k)$. Similar to GD however, only when all the local quantized gradients $\{Q_m(\boldsymbol{\theta}^k)\}$ are collected, the server can update the parameter $\boldsymbol{\theta}$.

In this context, the present paper puts forth a quantized gradient innovation method (as simple as QGD) that can *skip* communication in certain rounds. Specifically, in contrast to the server-to-worker downlink communication that can be performed simultaneously (e.g., by broadcasting $\boldsymbol{\theta}^k$), the server has to receive the workers' gradients sequentially to avoid interference from other workers, which leads to extra latency. For this reason, our focus here is on reducing the number of worker-to-server uplink communications, which we will also refer to as uploads. Our algorithm **L**azily **A**ggregated **Q**uantized gradient descent (**LAQ**) resembles (3), and it is given by

**LAQ iteration** $$\boldsymbol{\theta}^{k+1} = \boldsymbol{\theta}^k - \alpha\nabla^k \quad \text{with} \quad \nabla^k = \nabla^{k-1} + \sum_{m \in \mathcal{M}^k} \delta Q_m^k \tag{4}$$

where $\nabla^k$ is an approximate aggregated gradient that summarizes the parameter change at iteration $k$, and $\delta Q_m^k := Q_m(\boldsymbol{\theta}^k) - Q_m(\hat{\boldsymbol{\theta}}_m^{k-1})$ is the difference between two quantized gradients of $f_m$ at the current iterate $\boldsymbol{\theta}^k$ and the old copy $\hat{\boldsymbol{\theta}}_m^{k-1}$. With a judicious selection criterion that will be introduced later, $\mathcal{M}^k$ denotes the subset of workers whose local $\delta Q_m^k$ is uploaded in iteration $k$, while parameter iterates are given by $\hat{\boldsymbol{\theta}}_m^k := \boldsymbol{\theta}^k, \forall m \in \mathcal{M}^k$, and $\hat{\boldsymbol{\theta}}_m^k := \hat{\boldsymbol{\theta}}_m^{k-1}, \forall m \notin \mathcal{M}^k$. Instead of requesting fresh quantized gradient from every worker in (3), the trick is to obtain $\nabla^k$ by refining the previous aggregated gradient $\nabla^{k-1}$; that is, using only the new gradients from the *selected* workers in $\mathcal{M}^k$, while reusing the outdated gradients from the rest of workers. If $\nabla^{k-1}$ is stored in the server, this simple modification scales down the per-iteration communication rounds from QGD's $M$ to LAQ's $|\mathcal{M}^k|$. Throughout the paper, one round of communication means one worker's upload.

Compared to the existing quantization schemes, LAQ first quantizes the gradient innovation — the difference of current gradient and previous quantized gradient, and then skips the gradient communication — if the gradient innovation of a worker is not large enough, the communication of this worker is skipped. We will rigorously establish that LAQ achieves the same linear convergence as GD under the strongly convex assumption of the loss function. Numerical tests will demonstrate that our approach outperforms existing methods in terms of both communication bits and rounds.

**Notation**. Bold lowercase letters denote column vectors; $\|\mathbf{x}\|_2$ and $\|\mathbf{x}\|_\infty$ denote the $\ell_2$-norm and $\ell_\infty$-norm of $\mathbf{x}$, respectively; and $[\mathbf{x}]_i$ represents $i$-th entry of $\mathbf{x}$; while $\lfloor a \rceil$ denotes downward rounding of $a$; and $|\cdot|$ denotes the cardinality of the set or vector.

## 2 LAQ: Lazily aggregated quantized gradient

To reduce the communication overhead, two complementary stages are integrated in our algorithm design: 1) gradient innovation-based quantization; and 2) gradient innovation-based uploading or aggregation — giving the name **L**azily **A**ggregated **Q**uantized gradient (**LAQ**). The former reduces the number of bits per upload, while the latter cuts down the number of uploads, which together guarantee parsimonious communication. This section explains the principles of our two-stage design.

### 2.1 Gradient innovation-based quantization

Quantization limits the number of bits to represent a gradient vector during communication. Suppose we use $b$ bits to quantize each coordinate of the gradient vector in contrast to 32 bits as in most computers. With $\mathcal{Q}$ denoting the quantization operator, the quan-

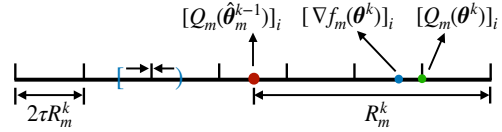

Figure 1: Quantization example ($b = 3$)

tized gradient for worker $m$ at iteration $k$ is $Q_m(\boldsymbol{\theta}^k) = \mathcal{Q}(\nabla f_m(\boldsymbol{\theta}^k), Q_m(\hat{\boldsymbol{\theta}}_m^{k-1}))$, which depends on the gradient $\nabla f_m(\boldsymbol{\theta}^k)$ and the previous quantization $Q_m(\hat{\boldsymbol{\theta}}_m^{k-1})$. The gradient is element-wise quantized by projecting to the closest point in a uniformly discretized grid. The grid is a $p$-dimensional hypercube which is centered at $Q_m(\hat{\boldsymbol{\theta}}_m^{k-1})$ with the radius $R_m^k = \|\nabla f_m(\boldsymbol{\theta}^k) - Q_m(\hat{\boldsymbol{\theta}}_m^{k-1})\|_\infty$. With $\tau := 1/(2^b - 1)$ defining the quantization granularity, the gradient innovation $f_m(\boldsymbol{\theta}^k) - Q_m(\hat{\boldsymbol{\theta}}_m^{k-1})$ can be quantized by $b$ bits per coordinate at worker $m$ as:

$$[q_m(\boldsymbol{\theta}^k)]_i = \left\lfloor \frac{[\nabla f_m(\boldsymbol{\theta}^k)]_i - [Q_m(\hat{\boldsymbol{\theta}}_m^{k-1})]_i + R_m^k}{2\tau R_m^k} + \frac{1}{2} \right\rfloor, \quad i = 1, \cdots, p \tag{5}$$

which is an integer within $[0, 2^b - 1]$, and thus can be encoded by $b$ bits. Note that adding $R_m^k$ in the numerator ensures the non-negativity of $[q_m(\boldsymbol{\theta}^k)]_i$, and adding $1/2$ in (5) guarantees rounding to the closest point. Hence, the *quantized* gradient innovation at worker $m$ is (with $\mathbf{1} := [1, \cdots, 1]^\top$)

$$\delta Q_m^k = Q_m(\boldsymbol{\theta}^k) - Q_m(\hat{\boldsymbol{\theta}}_m^{k-1}) = 2\tau R_m^k q_m(\boldsymbol{\theta}^k) - R_m^k \mathbf{1} : \quad \text{transmit} \quad R_m^k \quad \text{and} \quad q_m(\boldsymbol{\theta}^k) \tag{6}$$

which can be transmitted by $32 + bp$ bits (32 bits for $R_m^k$ and $bp$ bits for $q_m(\boldsymbol{\theta}^k)$) instead of the original $32p$ bits. With the outdated gradients $Q_m(\hat{\boldsymbol{\theta}}_m^{k-1})$ stored in the memory and $\tau$ known a priori, after receiving $\delta Q_m^k$ the server can recover the quantized gradient as $Q_m(\boldsymbol{\theta}^k) = Q_m(\hat{\boldsymbol{\theta}}_m^{k-1}) + \delta Q_m^k$.

Figure 1 gives an example for quantizing one coordinate of the gradient with $b = 3$ bits. The original value is quantized with 3 bits and $2^3 = 8$ values, each of which covers a range of length $2\tau R_m^k$ centered at itself. With $\varepsilon_m^k := \nabla f_m(\boldsymbol{\theta}^k) - Q_m(\boldsymbol{\theta}^k)$ denoting the local quantization error, it is clear that the quantization error is less than half of the length of the range that each value covers, namely, $\|\varepsilon_m^k\|_\infty \le \tau R_m^k$. The aggregated quantized gradient is $Q(\boldsymbol{\theta}^k) = \sum_{m \in \mathcal{M}} Q_m(\boldsymbol{\theta}^k)$, and the aggregated quantization error is $\varepsilon^k := \nabla f(\boldsymbol{\theta}^k) - Q(\boldsymbol{\theta}^k) = \sum_{m=1}^M \varepsilon_m^k$; that is, $Q(\boldsymbol{\theta}^k) = \nabla f(\boldsymbol{\theta}^k) - \varepsilon^k$.

## 2.2 Gradient innovation-based aggregation

The idea of lazy gradient aggregation is that if the difference of two consecutive locally quantized gradients is small, it is safe to skip the redundant gradient upload, and reuse the previous one at the server. In addition, we also ensure the server has a relatively "fresh" gradient for each worker by enforcing communication if any worker has not uploaded during the last $\bar{t}$ rounds. We set a clock $t_m$, $m \in \mathcal{M}$ for worker $m$ counting the number of iterations since last time it uploaded information. Equipped with the quantization and selection, our LAQ update takes the form as (4).

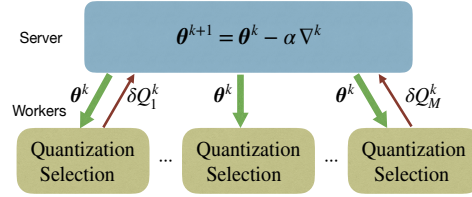

Figure 2: Distributed learning via LAQ

Now it only remains to design the selection criterion to decide which worker to upload the quantized gradient or its innovation. We propose the following communication criterion: worker $m \in \mathcal{M}$ skips the upload at iteration $k$, if it satisfies

$$\|Q_m(\hat{\boldsymbol{\theta}}_m^{k-1}) - Q_m(\boldsymbol{\theta}^k)\|_2^2 \le \frac{1}{\alpha^2 M^2} \sum_{d=1}^D \xi_d \|\boldsymbol{\theta}^{k+1-d} - \boldsymbol{\theta}^{k-d}\|_2^2 + 3\left(\|\varepsilon_m^k\|_2^2 + \|\hat{\varepsilon}_m^{k-1}\|_2^2\right); \tag{7a}$$

$$t_m \le \bar{t} \tag{7b}$$

where $D \le \bar{t}$ and $\{\xi_d\}_{d=1}^D$ are predetermined constants, $\varepsilon_m^k$ is the current quantization error, and $\hat{\varepsilon}_m^{k-1} = \nabla f_m(\hat{\boldsymbol{\theta}}_m^{k-1}) - Q_m(\hat{\boldsymbol{\theta}}_m^{k-1})$ is the error of the last uploaded quantized gradient. In next section we will prove the convergence and communication properties of LAQ under criterion (7).

## 2.3 LAQ algorithm development

In summary, as illustrated in Figure 2, LAQ can be implemented as follows. At iteration $k$, the server broadcasts the learning parameter to all workers. Each worker calculates the gradient, and then quantizes it to judge if it needs to upload the quantized gradient innovation $\delta Q_m^k$. Then the server updates the learning parameter after it receives the gradient innovation from the selected workers. The algorithm is summarized in Algorithm 2.

To make the difference between LAQ and GD clear, we re-write (4) as:

$$\boldsymbol{\theta}^{k+1} = \boldsymbol{\theta}^k - \alpha[\nabla Q(\boldsymbol{\theta}^k) + \sum_{m \in \mathcal{M}_c^k} (Q_m(\hat{\boldsymbol{\theta}}_m^{k-1}) - Q_m(\boldsymbol{\theta}^k))] \tag{8a}$$

$$= \boldsymbol{\theta}^k - \alpha[\nabla f(\boldsymbol{\theta}^k) - \varepsilon^k + \sum_{m \in \mathcal{M}_c^k} (Q_m(\hat{\boldsymbol{\theta}}_m^{k-1}) - Q_m(\boldsymbol{\theta}^k))] \tag{8b}$$

where $\mathcal{M}_c^k := \mathcal{M} \backslash \mathcal{M}^k$, is the subset of workers which skip communication with server at iteration $k$. Compared with the GD iteration in (2), the gradient employed here degrades due to the quantization error, $\varepsilon^k$ and the missed gradient innovation, $\sum_{m \in \mathcal{M}_c^k} (Q_m(\hat{\boldsymbol{\theta}}_m^{k-1}) - Q_m(\boldsymbol{\theta}^k))]$. It is clear that if large

**Algorithm 1 QGD**

1: **Input:** stepsize $\alpha > 0$, quantization bit $b$.
2: **Initialize:** $\boldsymbol{\theta}^k$.
3: **for** $k = 1, 2, \cdots, K$ **do**
4:     Server broadcasts $\boldsymbol{\theta}^k$ to all workers.
5:     **for** $m = 1, 2, \cdots, M$ **do**
6:         Worker $m$ computes $\nabla f_m(\boldsymbol{\theta}^k)$ and $Q_m(\boldsymbol{\theta}^k)$.
7:         Worker $m$ uploads $\delta Q_m^k$ via (6).
8:     **end for**
9:     Server updates $\boldsymbol{\theta}$ following (4) with $\mathcal{M}^k = \mathcal{M}$.
10: **end for**

**Algorithm 2 LAQ**

1: **Input:** stepsize $\alpha > 0$, $b$, $D$, $\{\xi_d\}_{d=1}^D$ and $\bar{t}$.
2: **Initialize:** $\boldsymbol{\theta}^k$, and $\{Q_m(\hat{\boldsymbol{\theta}}_m^0), t_m\}_{m \in \mathcal{M}}$.
3: **for** $k = 1, 2, \cdots, K$ **do**
4:     Server broadcasts $\boldsymbol{\theta}^k$ to all workers.
5:     **for** $m = 1, 2, \cdots, M$ **do**
6:         Worker $m$ computes $\nabla f_m(\boldsymbol{\theta}^k)$ and $Q_m(\boldsymbol{\theta}^k)$.
7:         **if** (7) holds for worker $m$ **then**
8:             Worker $m$ uploads nothing.
9:             Set $\hat{\boldsymbol{\theta}}_m^k = \hat{\boldsymbol{\theta}}_m^{k-1}$ and $t_m \leftarrow t_m + 1$.
10:        **else**
11:            Worker $m$ uploads $\delta Q_m^k$ via (6).
12:            Set $\hat{\boldsymbol{\theta}}_m^k = \boldsymbol{\theta}^k$, and $t_m = 0$.
13:        **end if**
14:    **end for**
15:    Server updates $\boldsymbol{\theta}$ according to (4).
16: **end for**

Table 1: A comparison of QGD and LAQ.

enough number of bits are used to quantize the gradient, and all $\{\xi_d\}_{d=1}^D$ are set 0 thus $\mathcal{M}^k := \mathcal{M}$, then LAQ reduces to GD. Thus, adjusting $b$ and $\{\xi_d\}_{d=1}^D$ directly influences the performance of LAQ.

The rationale behind selection criterion (7) lies in the judicious comparison between the descent amount of GD and that of LAQ. To compare the descent amount, we first establish the one step descent amount of both algorithms. For all the results in this paper, the following assumption holds.

**Assumption 1.** *The local gradient $\nabla f_m(\cdot)$ is $L_m$-Lipschitz continuous and the global gradient $\nabla f(\cdot)$ is $L$-Lipschitz continuous, i.e., there exist constants $L_m$ and $L$ such that*

$$\|\nabla f_m(\boldsymbol{\theta}_1) - \nabla f_m(\boldsymbol{\theta}_2)\|_2 \leq L_m\|\boldsymbol{\theta}_1 - \boldsymbol{\theta}_2\|_2, \; \forall \boldsymbol{\theta}_1, \boldsymbol{\theta}_2; \tag{9a}$$

$$\|\nabla f(\boldsymbol{\theta}_1) - \nabla f(\boldsymbol{\theta}_2)\|_2 \leq L\|\boldsymbol{\theta}_1 - \boldsymbol{\theta}_2\|_2, \; \forall \boldsymbol{\theta}_1, \boldsymbol{\theta}_2. \tag{9b}$$

Building upon Assumption 1, the next lemma describes the descent in objective by GD.

**Lemma 1.** *The gradient descent update yields following descent:*

$$f(\boldsymbol{\theta}^{k+1}) - f(\boldsymbol{\theta}^k) \leq \Delta_{GD}^k \tag{10}$$

*where $\Delta_{GD}^k := -(1 - \frac{\alpha L}{2})\alpha\|\nabla f(\boldsymbol{\theta}^k)\|_2^2$.*

The descent of LAQ distinguishes from that of GD due to the quantization and selection, which is specified in the following lemma.

**Lemma 2.** *The LAQ update yields following descent:*

$$f(\boldsymbol{\theta}^{k+1}) - f(\boldsymbol{\theta}^k) \leq \Delta_{LAQ}^k + \alpha\|\boldsymbol{\varepsilon}^k\|_2^2 \tag{11}$$

*where $\Delta_{LAQ}^k := -\frac{\alpha}{2}\|\nabla f(\boldsymbol{\theta}^k)\|_2^2 + \alpha\|\sum_{m \in \mathcal{M}_c^k}(Q_m(\hat{\boldsymbol{\theta}}_m^{k-1}) - Q_m(\boldsymbol{\theta}^k))\|_2^2 + (\frac{L}{2} - \frac{1}{2\alpha})\|\boldsymbol{\theta}^{k+1} - \boldsymbol{\theta}^k\|_2^2$.*

In lazy aggregation, we consider only $\Delta_{LAQ}^k$ with the quantization error in (11) ignored. Rigorous theorem showing the property of LAQ taking into account the quantization error will be established in next section.

The following part shows the intuition for criterion (7a), which is not mathematically strict but provides the intuition. The lazy aggregation mechanism selects the quantized gradient innovation by judging its contribution to decreasing the loss function. LAQ is expected to be more communication-efficient than GD, that is, each upload results in more descent, which translates to:

$$\frac{\Delta_{LAQ}^k}{|\mathcal{M}^k|} \leq \frac{\Delta_{GD}^k}{M}. \tag{12}$$

which is tantamount to (see the derivations in the supplementary materials)

$$\|(Q_m(\hat{\boldsymbol{\theta}}_m^{k-1}) - Q_m(\boldsymbol{\theta}^k)\|_2^2 \leq \|\nabla f(\boldsymbol{\theta}^k)\|_2^2/(2M^2), \; \forall m \in \mathcal{M}_c^k. \tag{13}$$

However, for each worker to check (73) locally is impossible because the fully aggregated gradient $\nabla f(\boldsymbol{\theta}^k)$ is required, which is exactly what we want to avoid. Moreover, it does not make sense to reduce uploads if the fully aggregated gradient has been obtained. Therefore, we bypass directly calculating $\|\nabla f(\boldsymbol{\theta}^k)\|_2^2$ using its approximation below.

$$\|\nabla f(\boldsymbol{\theta}^k)\|_2^2 \approx \frac{2}{\alpha^2} \sum_{k=1}^{D} \xi_d \|\boldsymbol{\theta}^{k+1-d} - \boldsymbol{\theta}^{k-d}\|_2^2 \tag{14}$$

where $\{\xi_d\}_{d=1}^D$ are constants. The fundamental reason why (74) holds is that $\nabla f(\boldsymbol{\theta}^k)$ can be approximated by weighted previous gradients or parameter differences since $f(\cdot)$ is $L$-smooth. Combining (73) and (74) leads to our communication criterion (7a) with quantization error ignored.

We conclude this section by a comparison between LAQ and error-feedback (quantized) schemes.

**Comparison with error-feedback schemes**. Our LAQ approach is related to the error-feedback schemes, e.g., [3, 12, 24, 26, 27, 32]. Both lines of approaches accumulate either errors or delayed innovation incurred by communication reduction (e.g., quantization, sparsification, or skipping), and upload them in the next communication round. However, the error-feedback schemes skip communicating certain entries of the gradient, yet communicate with all workers. LAQ skips communicating with certain workers, but communicates all (quantized) entries. The two methods are not mutually exclusive, and can be used jointly.

# 3 Convergence and communication analysis

Our subsequent convergence analysis of LAQ relies on the following assumption on $f(\boldsymbol{\theta})$:

**Assumption 2.** *The function $f(\cdot)$ is $\mu$-strongly convex, e.g., there exists a constant $\mu > 0$ such that*

$$f(\boldsymbol{\theta}_1) - f(\boldsymbol{\theta}_2) \geq \langle \nabla f(\boldsymbol{\theta}_2), \boldsymbol{\theta}_1 - \boldsymbol{\theta}_2 \rangle + \frac{\mu}{2} \|\boldsymbol{\theta}_1 - \boldsymbol{\theta}_2\|_2^2, \quad \forall \boldsymbol{\theta}_1, \boldsymbol{\theta}_2. \tag{15}$$

With $\boldsymbol{\theta}^*$ denoting the optimal solution of (1), we define Lyapunov function of LAQ as:

$$\mathbb{V}(\boldsymbol{\theta}^k) = f(\boldsymbol{\theta}^k) - f(\boldsymbol{\theta}^*) + \sum_{d=1}^{D} \sum_{j=d}^{D} \frac{\xi_j}{\alpha} \|\boldsymbol{\theta}^{k+1-d} - \boldsymbol{\theta}^{k-d}\|_2^2 \tag{16}$$

The design of Lyapunov function $\mathbb{V}(\boldsymbol{\theta})$ is coupled with the communication rule (7a) that contains parameter difference term. Intuitively, if no communication is being skipped at current iteration, LAQ behaves like GD that decreases the objective residual in $\mathbb{V}(\boldsymbol{\theta})$; if certain uploads are skipped, LAQ's rule (7a) guarantees the error of using stale gradients comparable to the parameter difference in $\mathbb{V}(\boldsymbol{\theta})$ to ensure its descending. The following lemma captures the progress of the Lyapunov function.

**Lemma 3.** *Under Assumptions 1 and 2, if the stepsize $\alpha$ and the parameters $\{\xi_d\}_{d=1}^D$ are selected as (with any $0 < \rho_1 < 1$ and $\rho_2 > 0$)*

$$\sum_{d=1}^{D} \xi_d \leq \min \left\{ \frac{1 - \rho_1}{4(1 + \rho_2)}, \frac{1}{2(1 + \rho_2^{-1})} \right\} \tag{17a}$$

$$\alpha \leq \min \left\{ \frac{2}{L} \left( \frac{1 - \rho_1}{4(1 + \rho_2)} - \sum_{d=1}^{D} \xi_d \right), \frac{2}{L} \left( \frac{1}{2(1 + \rho_2^{-1})} - \sum_{d=1}^{D} \xi_d \right) \right\} \tag{17b}$$

*then the Lyapunov function follows*

$$\mathbb{V}(\boldsymbol{\theta}^{k+1}) \leq \sigma_1 \mathbb{V}(\boldsymbol{\theta}^k) + B \left[ \|\boldsymbol{\varepsilon}^k\|_2^2 + \sum_{m \in \mathcal{M}_c^k} \left( \|\boldsymbol{\varepsilon}_m^k\|_2^2 + \|\hat{\boldsymbol{\varepsilon}}_m^{k-1}\|_2^2 \right) \right] \tag{18}$$

*where constants $0 < \sigma_1 < 1$ and $B > 0$ depend on $\alpha$ and $\{\xi_d\}$; see details in supplementary materials.*

For the tight analysis, (17) appear to be involved, but it admits simple choices. For example, when we choose $\rho_1 = 1/2$ and $\rho_2 = 1$, respectively, then $\xi_1 = \xi_2 = \cdots \xi_D = \frac{1}{16D}$ and $\alpha = \frac{1}{8L}$ satisfy (17).

If the quantization error in (18) is null, Lemma 3 readily implies that the Lyapunov function enjoys a linear convergence rate. In the following, we will demonstrate that under certain conditions, the LAQ algorithm can still guarantee linear convergence even if we consider the the quantization error.

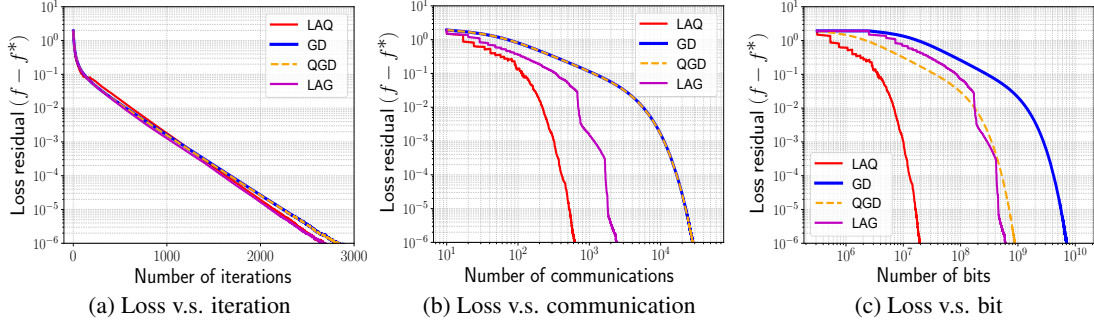

(a) Loss v.s. iteration　　　　(b) Loss v.s. communication　　　　(c) Loss v.s. bit

Figure 4: Convergence of the loss function (logistic regression)

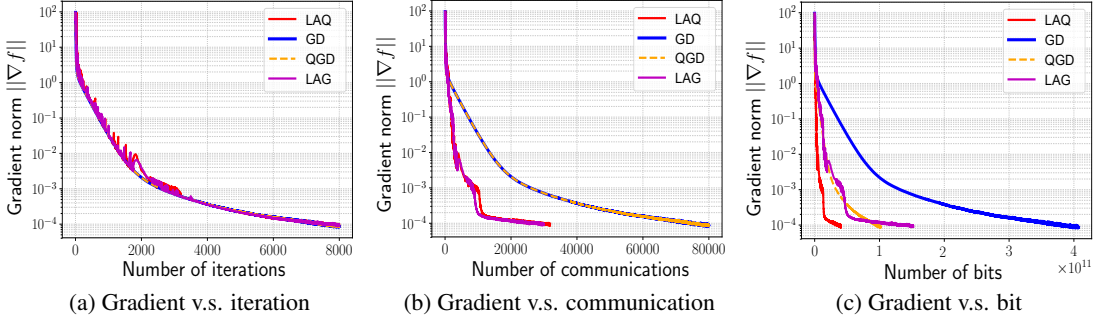

(a) Gradient v.s. iteration　　　　(b) Gradient v.s. communication　　　　(c) Gradient v.s. bit

Figure 5: Convergence of gradient norm (neural network)

**Theorem 1.** *Under the same assumptions and the parameters in Lemma 3, Lyapunov function and the quantization error converge at a linear rate; that is, there exists a constant $\sigma_2 \in (0,1)$ such that*

$$\mathbb{V}(\boldsymbol{\theta}^k) \leq \sigma_2^k P; \tag{19a}$$

$$\|\boldsymbol{\varepsilon}_m^k\|_\infty^2 \leq \tau^2 \sigma_2^k P, \; \forall m \in \mathcal{M}. \tag{19b}$$

*where $P$ is a constant depending on the parameters in (17); see details in supplementary materials.*

From the definition of Lyapunov function, it is clear that $f(\boldsymbol{\theta}^k) - f(\boldsymbol{\theta}^*) \leq \mathbb{V}(\boldsymbol{\theta}^k) \leq \sigma_2^k \mathbb{V}^0$ — the risk error $f(\boldsymbol{\theta}^k) - f(\boldsymbol{\theta}^*)$ converges linearly. The $L$-smoothness results in $\|\nabla f(\boldsymbol{\theta}^k)\|_2^2 \leq 2L[f(\boldsymbol{\theta}^k - f(\boldsymbol{\theta}^*)] \leq 2L\sigma_2^k \mathbb{V}^0$ — the gradient norm $\|\nabla f(\boldsymbol{\theta}^k)\|_2^2$ converges linearly. Similarly, the $\mu$-strong convexity implies $\|\boldsymbol{\theta}^k - \boldsymbol{\theta}^*\|_2^2 \leq \frac{2}{\mu}[f(\boldsymbol{\theta}^k - f(\boldsymbol{\theta}^*)] \leq \frac{2}{\mu}\sigma_2^k \mathbb{V}^0$ — $\|\boldsymbol{\theta}^k - \boldsymbol{\theta}^*\|_2^2$ also converges linearly.

Compared to the previous analysis for LAG [6], the analysis for LAQ is more involved, since it needs to deal with not only outdated but also quantized (inexact) gradients. This modification deteriorates the monotonic property of the Lyapunov function in (18), which is the building block of analysis in [6]. We tackle this issue by i) considering the outdated gradient in the quantization (6); and, ii) incorporating quantization error in the new selection criterion (7). As a result, Theorem 1 demonstrates that LAQ is able to keep the linear convergence rate even with the presence of the quantization error. This is because the properly controlled quantization error also converges at a linear rate; see the illustration in Figure 3.

**Proposition 1.** *Under Assumption 1, if we choose the constants $\{\xi_d\}_{d=1}^D$ satisfying $\xi_1 \geq \xi_2 \geq \cdots \geq \xi_D$ and define $d_m$, $m \in \mathcal{M}$ as:*

$$d_m := \max_d \left\{ d \mid L_m^2 \leq \xi_d/(3\alpha^2 M^2 D), d \in \{1, 2, \cdots, D\} \right\} \tag{20}$$

*then, worker $m$ has at most $k/(d_m + 1)$ communications with the server until the $k$-th iteration.*

This proposition implies that the smoothness of the local loss function determines the communication intensity of the local worker.

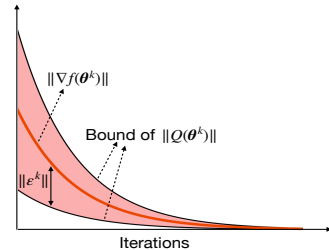

Figure 3: Gradient norm decay

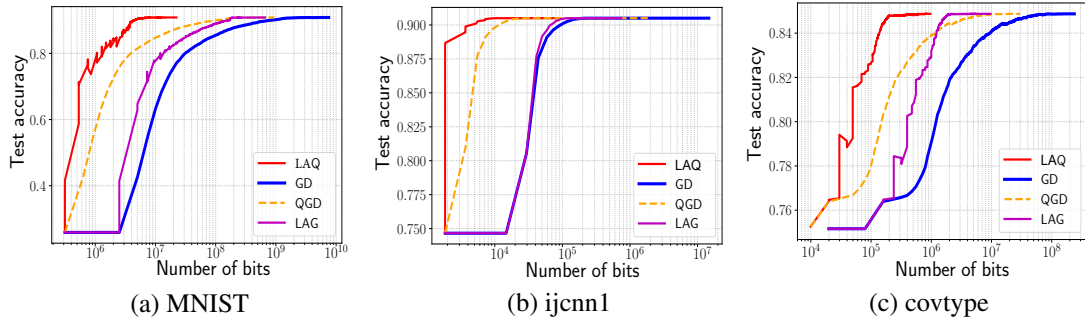

Figure 6: Test accuracies on three different datasets

(a) MNIST    (b) ijcnn1    (c) covtype

# 4 Numerical tests and conclusions

To validate our performance analysis and verify its communication savings in practical machine learning problems, we evaluate the performance of the algorithm for the regularized logistic regression which is strongly convex, and the neural network which is nonconvex. The dataset we use is MNIST [15], which are uniformly distributed across $M = 10$ workers. In the experiments, we set $D = 10$, $\xi_1 = \xi_2 = \cdots, \xi_D = 0.8/D$, $\bar{t} = 100$; see the detailed setup in the supplementary materials. To benchmark LAQ, we compare it with two classes of algorithms, gradient-based algorithms and minibatch stochastic gradient-based algorithms — corresponding to the following two tests.

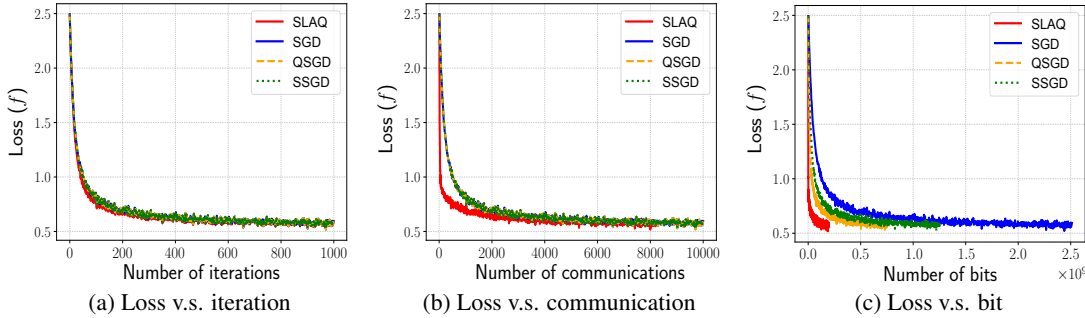

(a) Loss v.s. iteration    (b) Loss v.s. communication    (c) Loss v.s. bit

Figure 7: Convergence of loss function (logistic regression)

**Gradient-based tests.** We consider GD, QGD [18] and lazily aggregated gradient (LAG) [6]. The number of bits per coordinate is set as $b = 3$ for logistic regression and 8 for neural network, respectively. Stepsize is set as $\alpha = 0.02$ for both algorithms. Figure 4 shows the objective convergence for the logistic regression task. Clearly, Figure 4(a) verifies Theorem 1, e.g., the linear convergence rate under strongly convex loss function. As shown in Figure 4(b), LAQ requires fewer number of communication rounds than GD and QGD thanks to our selection rule, but more rounds than LAG due to the gradient quantization. Nevertheless, the total number of transmitted bits of LAQ is significantly smaller than that of LAG, as demonstrated in Figure 4(c). For neural network model, Figure 5 reports the convergence of gradient norm, where LAQ also shows competitive performance

| | Algorithm | Iteration # | Communication # | Bit # | Accuracy |
|---|---|---|---|---|---|
| **LAQ** | logistic | **2673** | **620** | $\mathbf{1.95 \times 10^7}$ | **0.9082** |
| | neural network | **8000** | **31845** | $\mathbf{4.05 \times 10^{10}}$ | **0.9433** |
| GD | logistic | 2820 | 28200 | $7.08 \times 10^9$ | 0.9082 |
| | neural network | 8000 | 80000 | $4.07 \times 10^{11}$ | 0.9433 |
| QGD | logistic | 2805 | 28050 | $8.81 \times 10^8$ | 0.9082 |
| | neural network | 8000 | 80000 | $1.02 \times 10^{11}$ | 0.9433 |
| LAG | logistic | 2659 | 2382 | $5.98 \times 10^8$ | 0.9082 |
| | neural network | 8000 | 29916 | $1.52 \times 10^{11}$ | 0.9433 |

Table 2: Comparison of gradient-based algorithms. For logistic regression, all algorithms terminate when loss residual reaches $10^{-6}$; for neural network, all algorithms run a fixed number of iterations.

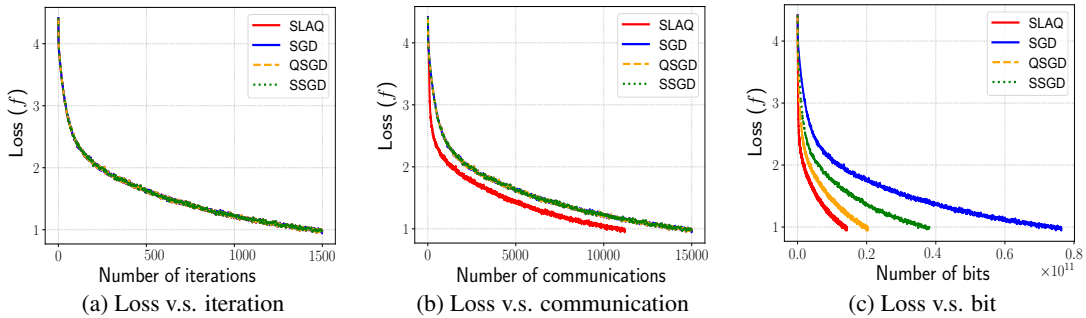

Figure 8: Convergence of loss function (neural network)

for nonconvex problem. Similar to the results for logistic model, LAQ requires the fewest number of bits. Table 2 summarizes the number of iterations, uploads and bits needed to reach a given accuracy.

Figure 6 exhibits the test accuracy of above compared algorithms on three commonly used datasets, MNIST, ijcnn1 and covtype. Applied to all these datasets, LAQ saves transmitted bits and meanwhile maintains the same accuracy.

**Stochastic gradient-based tests.** We test stochastic gradient descent (SGD), quantized stochastic gradient descent (QSGD) [2], sparsified stochastic gradient descent (SSGD) [30], and the stochastic version of LAQ abbreviated as SLAQ. The mini-batch size is $500$, $\alpha = 0.008$, and the number of bits per coordinate is set as $b = 3$ for logistic regression and $8$ for neural network. As shown in Figures 7 and 8, SLAQ incurs the lowest number of communication rounds and bits. In this stochastic gradient test, although the communication reduction of SLAQ is not as significant as LAQ compared with gradient based algorithms, SLAQ still outperforms the state-of-the-art algorithms, e.g., QSGD and SSGD. The results are summarized in Table 3. More results under different number of bits and the level of heterogeneity are reported in the supplementary materials.

| Algorithm | | Iteration # | Communication # | Bit # | Accuracy |
|---|---|---|---|---|---|
| **SLAQ** | logistic | **1000** | **8255** | $\mathbf{1.94 \times 10^8}$ | **0.9018** |
| | neural network | **1500** | **11192** | $\mathbf{1.42 \times 10^{10}}$ | **0.9107** |
| SGD | logistic | 1000 | 10000 | $2.51 \times 10^9$ | 0.9021 |
| | neural network | 1500 | 15000 | $7.63 \times 10^{10}$ | 0.9100 |
| QSGD | logistic | 1000 | 10000 | $7.51 \times 10^8$ | 0.9021 |
| | neural network | 1500 | 15000 | $2.03 \times 10^{10}$ | 0.9100 |
| SSGD | logistic | 1000 | 10000 | $1.26 \times 10^9$ | 0.9013 |
| | neural network | 1500 | 15000 | $3.82 \times 10^{10}$ | 0.9104 |

Table 3: Performance comparison of mini-batch stochastic gradient-based algorithms.

This paper studied the communication-efficient distributed learning problem, and proposed LAQ that simultaneously quantizes and skips the communication based on gradient innovation. Compared to the original GD method, linear convergence rate is still maintained for strongly convex loss function. This is remarkable since LAQ saves both communication bits and rounds significantly. Numerical tests using (strongly convex) regularized logistic regression and (nonconvex) neural network models demonstrate the advantages of LAQ over existing popular approaches.

### Acknowledgments

This work by J. Sun and Z. Yang is supported in part by the Shenzhen Committee on Science and Innovations under Grant GJHZ20180411143603361, in part by the Department of Science and Technology of Guangdong Province under Grant 2018A050506003, and in part by the Natural Science Foundation of China under Grant 61873118. The work by J. Sun is also supported by China Scholarship Council. The work by G. Giannakis is supported in part by NSF 1500713, and 1711471.

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
