[Supplementary Material · neurips2019_supp.pdf]

# Supplementary materials for "Communication-Efficient Distributed Learning via Lazily Aggregated Quantized Gradients"

## A  Proof of Lemma 2

With the LAQ update, we have:

$$f(\boldsymbol{\theta}^{k+1}) - f(\boldsymbol{\theta}^k) \leq \left\langle \nabla f(\boldsymbol{\theta}^k), -\alpha[\nabla f(\boldsymbol{\theta}^k) - \boldsymbol{\varepsilon}^k + \sum_{m \in \mathcal{M}_c^k} (Q_m(\hat{\boldsymbol{\theta}}_m^{k-1}) - Q_m(\boldsymbol{\theta}^k))] \right\rangle + \frac{L}{2} \|\boldsymbol{\theta}^{k+1} - \boldsymbol{\theta}^k\|_2^2$$

$$\leq -\alpha \|\nabla f(\boldsymbol{\theta}^k)\|_2^2 + \alpha \left\langle \nabla f(\boldsymbol{\theta}^k), \boldsymbol{\varepsilon}^k - \sum_{m \in \mathcal{M}_c^k} (Q_m(\hat{\boldsymbol{\theta}}_m^{k-1}) - Q_m(\boldsymbol{\theta}^k)) \right\rangle + \frac{L}{2} \|\boldsymbol{\theta}^{k+1} - \boldsymbol{\theta}^k\|_2^2$$

$$= -\alpha \|\nabla f(\boldsymbol{\theta}^k)\|_2^2 + \frac{\alpha}{2} [\|\nabla f(\boldsymbol{\theta}^k)\|_2^2 + \|\boldsymbol{\varepsilon}^k - \sum_{m \in \mathcal{M}_c^k} (Q_m(\hat{\boldsymbol{\theta}}_m^{k-1}) - Q_m(\boldsymbol{\theta}^k))\|_2^2 - \frac{\|\boldsymbol{\theta}^{k+1} - \boldsymbol{\theta}^k\|_2^2}{\alpha^2}] + \frac{L}{2} \|\boldsymbol{\theta}^{k+1} - \boldsymbol{\theta}^k\|_2^2$$

$$\leq -\frac{\alpha}{2} \|\nabla f(\boldsymbol{\theta}^k)\|_2^2 + \alpha \| \sum_{m \in \mathcal{M}_c^k} (Q_m(\hat{\boldsymbol{\theta}}_m^{k-1}) - Q_m(\boldsymbol{\theta}^k))\|_2^2 + (\frac{L}{2} - \frac{1}{2\alpha}) \|\boldsymbol{\theta}^{k+1} - \boldsymbol{\theta}^k\|_2^2 + \alpha \|\boldsymbol{\varepsilon}^k\|_2^2$$

where the second equality follows from: $\langle \boldsymbol{a}, \boldsymbol{b} \rangle = \frac{1}{2}(\|\boldsymbol{a}\|^2 + \|\boldsymbol{b}\|^2 - \|\boldsymbol{a} - \boldsymbol{b}\|^2)$ and the last inequality is resulted from: $\|\sum_{i=1}^n \boldsymbol{a}_i\|_2^2 \leq n \sum_{i=1}^n \|\boldsymbol{a}_i\|^2$.

## B  Proof of Lemma 3

With Assumption 1, under the LAQ we have:

$$f(\boldsymbol{\theta}^{k+1}) - f(\boldsymbol{\theta}^k) \leq \left\langle \nabla f(\boldsymbol{\theta}^k), -\alpha[Q(\boldsymbol{\theta}^k) + \sum_{m \in \mathcal{M}_c^k} (Q_m(\hat{\boldsymbol{\theta}}_m^{k-1}) - Q_m(\boldsymbol{\theta}^k))] \right\rangle + \frac{L}{2} \|\boldsymbol{\theta}^{k+1} - \boldsymbol{\theta}^k\|_2^2$$

$$\leq -\left\langle \nabla f(\boldsymbol{\theta}^k), \alpha Q(\boldsymbol{\theta}^k) \right\rangle + \frac{\alpha}{2} \|\nabla f(\boldsymbol{\theta}^k)\|_2^2 + \frac{\alpha}{2} \| \sum_{m \in \mathcal{M}_c^k} Q_m(\hat{\boldsymbol{\theta}}_m^{k-1}) - Q_m(\boldsymbol{\theta}^k)\|_2^2 + \frac{L}{2} \|\boldsymbol{\theta}^{k+1} - \boldsymbol{\theta}^k\|_2^2$$

For the ease of expression, we define $\beta_d := \frac{1}{\alpha} \sum_{j=d}^D \xi_j$, $d = 1, 2, \cdots, D$. Then the Lyapunov function defined in (16) can be written as

$$\mathbb{V}(\boldsymbol{\theta}^k) = f(\boldsymbol{\theta}^k) - f(\boldsymbol{\theta}^*) + \sum_{d=1}^D \beta_d \|\boldsymbol{\theta}^{k+1-d} - \boldsymbol{\theta}^{k-d}\|_2^2. \tag{21}$$

Thus, we have
$$\mathbb{V}(\boldsymbol{\theta}^{k+1}) - \mathbb{V}(\boldsymbol{\theta}^k)$$

$$\leq -\alpha \left\langle \nabla f(\boldsymbol{\theta}^k), Q(\boldsymbol{\theta}^k) \right\rangle + \frac{\alpha}{2} \|\nabla f(\boldsymbol{\theta}^k)\|_2^2 + \frac{\alpha}{2} \| \sum_{m \in \mathcal{M}_c^k} Q_m(\hat{\boldsymbol{\theta}}_m^{k-1}) - Q_m(\boldsymbol{\theta}^k)\|_2^2 + (\frac{L}{2} + \beta_1) \|\boldsymbol{\theta}^{k+1} - \boldsymbol{\theta}^k\|_2^2$$

$$+ \sum_{d=1}^{D-1} (\beta_{d+1} - \beta_d) \|\boldsymbol{\theta}^{k+1-d} - \boldsymbol{\theta}^{k-d}\|_2^2 - \beta_D \|\boldsymbol{\theta}^{k+1-D} - \boldsymbol{\theta}^{k-D}\|_2^2$$

$$\leq -\alpha \left\langle \nabla f(\boldsymbol{\theta}^k), Q(\boldsymbol{\theta}^k) \right\rangle + \frac{\alpha}{2} \|\nabla f(\boldsymbol{\theta}^k)\|_2^2 + [\frac{\alpha}{2} + (\frac{L}{2} + \beta_1)(1 + \rho_2^{-1})\alpha^2] \| \sum_{m \in \mathcal{M}_c^k} Q_m(\hat{\boldsymbol{\theta}}_m^{k-1}) - Q_m(\boldsymbol{\theta}^k)\|_2^2$$

$$+ (\frac{L}{2} + \beta_1)(1 + \rho_2)\alpha^2 \|Q(\boldsymbol{\theta}^k)\|_2^2 + \sum_{d=1}^{D-1} (\beta_{d+1} - \beta_d) \|\boldsymbol{\theta}^{k+1-d} - \boldsymbol{\theta}^{k-d}\|_2^2 - \beta_D \|\boldsymbol{\theta}^{k+1-D} - \boldsymbol{\theta}^{k-D}\|_2^2$$

$$\leq -\alpha \left\langle \nabla f(\boldsymbol{\theta}^k), Q(\boldsymbol{\theta}^k) \right\rangle + \frac{\alpha}{2} \|\nabla f(\boldsymbol{\theta}^k)\|_2^2 + [\frac{\alpha}{2} + (\frac{L}{2} + \beta_1)(1 + \rho_2^{-1})\alpha^2] \frac{1}{\alpha^2} \sum_{d=1}^D \xi_d \|\boldsymbol{\theta}^{k+1-d} - \boldsymbol{\theta}^{k-d}\|_2^2$$

$$+ (\frac{L}{2} + \beta_1)(1 + \rho_2)\alpha^2 \|Q(\boldsymbol{\theta}^k)\|_2^2 + \sum_{d=1}^{D-1} (\beta_{d+1} - \beta_d) \|\boldsymbol{\theta}^{k+1-d} - \boldsymbol{\theta}^{k-d}\|_2^2 - \beta_D \|\boldsymbol{\theta}^{k+1-D} - \boldsymbol{\theta}^{k-D}\|_2^2$$

$$+ [\frac{3\alpha}{2} + (\frac{3L}{2} + 3\beta_1)(1 + \rho_2^{-1})\alpha^2] M(\|\varepsilon_m^k\|_2^2 + \|\hat{\varepsilon}_m^{k-1}\|_2^2)$$

$$\tag{22}$$

where the second inequality follows from Young's Equality: $\|\boldsymbol{a} + \boldsymbol{b}\|_2^2 \le (1+\rho)\|\boldsymbol{a}\|_2^2 + (1+\rho^{-1})\|\boldsymbol{b}\|_2^2$. The last inequality is resulted from

$$
\begin{aligned}
\| \sum_{m \in \mathcal{M}_c^k} Q_m(\hat{\boldsymbol{\theta}}_m^{k-1}) - Q_m(\boldsymbol{\theta}^k)\|_2^2 &\le |\mathcal{M}_c^k| \sum_{m \in \mathcal{M}_c^k} \|Q_m(\hat{\boldsymbol{\theta}}_m^{k-1}) - Q_m(\boldsymbol{\theta}^k)\|_2^2 \\
&\le \frac{|\mathcal{M}_c^k|^2}{\alpha^2 |\mathcal{M}|^2} \sum_{d=1}^{D} \xi_d \|\boldsymbol{\theta}^{k+1-d} - \boldsymbol{\theta}^{k-d}\|_2^2 + 3|\mathcal{M}_c^k| \sum_{m \in \mathcal{M}_c^k} (\|\varepsilon_m^k\|_2^2 + \|\hat{\varepsilon}_m^{k-1}\|_2^2) \\
&\le \frac{1}{\alpha^2} \sum_{d=1}^{D} \xi_d \|\boldsymbol{\theta}^{k+1-d} - \boldsymbol{\theta}^{k-d}\|_2^2 + 3M \sum_{m \in \mathcal{M}_c^k} (\|\varepsilon_m^k\|_2^2 + \|\hat{\varepsilon}_m^{k-1}\|_2^2)
\end{aligned}
\tag{23}
$$

where the second inequality follows from (7a). Substituting $Q(\boldsymbol{\theta}^k) = \nabla f(\boldsymbol{\theta}^k) - \varepsilon^k$ into (22) gives

$$
\begin{aligned}
&\mathbb{V}(\boldsymbol{\theta}^{k+1}) - \mathbb{V}(\boldsymbol{\theta}^k) \\
&\le -\frac{\alpha}{2} \|\nabla f(\boldsymbol{\theta}^k)\|_2^2 + \alpha \left\langle \nabla f(\boldsymbol{\theta}^k), \varepsilon^k \right\rangle + [\frac{\alpha}{2} + (\frac{L}{2} + \beta_1)(1 + \rho_2^{-1})\alpha^2] \frac{1}{\alpha^2} \sum_{d=1}^{D} \xi_d \|\boldsymbol{\theta}^{k+1-d} - \boldsymbol{\theta}^{k-d}\|_2^2 \\
&+ (\frac{L}{2} + \beta_1)(1 + \rho_2)\alpha^2 \|\nabla f(\boldsymbol{\theta}^k) - \varepsilon^k\|_2^2 + \sum_{d=1}^{D-1} (\beta_{d+1} - \beta_d) \|\boldsymbol{\theta}^{k+1-d} - \boldsymbol{\theta}^{k-d}\|_2^2 \\
&- \beta_D \|\boldsymbol{\theta}^{k+1-D} - \boldsymbol{\theta}^{k-D}\|_2^2 + (\frac{3L}{2} + 3\beta_1)(1 + \rho_2^{-1})\alpha^2]M \sum_{m \in \mathcal{M}_c^k} (\|\varepsilon_m^k\|_2^2 + \|\hat{\varepsilon}_m^{k-1}\|_2^2) \\
&\le [(-\frac{1}{2} + \frac{1}{2}\rho_1)\alpha + (L + 2\beta_1)(1 + \rho_2)\alpha^2] \|\nabla f(\boldsymbol{\theta}^k)\|_2^2 + \{[\frac{\alpha}{2} + (\frac{L}{2} + \beta_1)(1 + \rho_2^{-1})\alpha^2] \frac{\xi_D}{\alpha^2} - \beta_D\} \\
&\|\boldsymbol{\theta}^{k+1-D} - \boldsymbol{\theta}^{k-D}\|_2^2 + \sum_{d=1}^{D-1} [\frac{\alpha}{2} + (\frac{L}{2} + \beta_1)(1 + \rho_2^{-1})\alpha^2] \frac{\xi_d}{\alpha^2} + \beta_{d+1} - \beta_d\} \|\boldsymbol{\theta}^{k+1-d} - \boldsymbol{\theta}^{k-d}\|_2^2 \\
&+ [\frac{3\alpha}{2} + (\frac{3L}{2} + 3\beta_1)(1 + \rho_2^{-1})\alpha^2]M \sum_{m \in \mathcal{M}_c^k} (\|\varepsilon_m^k\|_2^2 + \|\hat{\varepsilon}_m^{k-1}\|_2^2) \\
&+ [\frac{1}{2\rho_1}\alpha + (L + 2\beta_1)(1 + \rho_2)\alpha^2] \|\varepsilon^k\|_2^2
\end{aligned}
\tag{24}
$$

where the second inequality is the consequence of

$$
\left\langle \nabla f(\boldsymbol{\theta}^k), \varepsilon^k \right\rangle \le \frac{1}{2}\rho_1 \|\nabla f(\boldsymbol{\theta}^k)\|_2^2 + \frac{1}{2\rho_1} \|\varepsilon^k\|_2^2.
\tag{25}
$$

The following conditions are sufficient to guarantee the first three terms in (24) are non-positive.

$$
(-\frac{1}{2} + \frac{1}{2}\rho_1)\alpha + (L + 2\beta_1)(1 + \rho_2)\alpha^2 \le 0;
\tag{26a}
$$

$$
[\frac{\alpha}{2} + (\frac{L}{2} + \beta_1)(1 + \rho_2^{-1})\alpha^2] \frac{\xi_d}{\alpha^2} + \beta_{d+1} - \beta_d \le 0, \ \forall d \in \{1, 2, \cdots, D-1\};
\tag{26b}
$$

$$
[\frac{\alpha}{2} + (\frac{L}{2} + \beta_1)(1 + \rho_2^{-1})\alpha^2] \frac{\xi_D}{\alpha^2} - \beta_D \le 0.
\tag{26c}
$$

By simple manipulation after replacing $\beta_d$ by $\frac{1}{\alpha} \sum_{j=d}^{D} \xi_j$ in (26), we attain (17).

Assumption 2 indicates $f(\cdot)$ satisfies the PL condition:

$$
2\mu(f(\boldsymbol{\theta}^k) - f(\boldsymbol{\theta}^*)) \le \|\nabla f(\boldsymbol{\theta}^k)\|_2^2.
\tag{27}
$$

Let $c$ be defined as

$$
c = \min_d \left\{ \mu(1 - \rho_1)\alpha - 2\mu(L + 2\beta_1)(1 + \rho_2)\alpha^2, 1 - \frac{[\frac{\alpha}{2} + (\frac{L}{2} + \beta_1)(1 + \rho_2^{-1})\alpha^2]\xi_D}{\alpha^2 \beta_D}, \right.
$$
$$
\left. 1 - \frac{\beta_{d+1}}{\beta_d} - \frac{[\frac{\alpha}{2} + (\frac{L}{2} + \beta_1)(1 + \rho_2^{-1})\alpha^2]\xi_d}{\alpha^2 \beta_d} \right\}.
\tag{28}
$$

Then,

$$\mathbb{V}(\boldsymbol{\theta}^{k+1})-\mathbb{V}(\boldsymbol{\theta}^k) \leq -c[f(\boldsymbol{\theta}^k)-f(\boldsymbol{\theta}^*)+\sum_{d=1}^{D}\beta_d\|\boldsymbol{\theta}^{k+1-d}-\boldsymbol{\theta}^{k-d}\|_2^2]+B[\|\boldsymbol{\varepsilon}^k\|_2^2+\sum_{m\in\mathcal{M}_c^k}(\|\boldsymbol{\varepsilon}_m^k\|_2^2+\|\hat{\boldsymbol{\varepsilon}}_m^{k-1}\|_2^2)],$$

(29)

i.e.,

$$\mathbb{V}(\boldsymbol{\theta}^{k+1}) \leq \sigma_1\mathbb{V}(\boldsymbol{\theta}^k) + B[\|\boldsymbol{\varepsilon}^k\|_2^2 + \sum_{m\in\mathcal{M}_c^k}(\|\boldsymbol{\varepsilon}_m^k\|_2^2+\|\hat{\boldsymbol{\varepsilon}}_m^{k-1}\|_2^2)],$$

(30)

where $\sigma_1 = 1 - c$, $B = \max\{\frac{1}{2\rho_1}\alpha + (L + 2\beta_1)(1+\rho_2)\alpha^2, [\frac{3\alpha}{2} + (\frac{3L}{2}+3\beta_1)(1+\rho_2^{-1})\alpha^2]M\}$.

## C   Proof of Theorem 1

We can first prove that there exist constants $\sigma_2 \in (0,1)$ and $B_1 > 0$ such that,

$$\mathbb{V}(\boldsymbol{\theta}^k) \leq \sigma_2^k \mathbb{V}^0;$$

(31a)

$$\|\boldsymbol{\varepsilon}_m^k\|_\infty^2 \leq B_1^2\tau^2\sigma_2^k\mathbb{V}^0, \ \forall m \in \mathcal{M}.$$

(31b)

where $\mathbb{V}^0$ is a constant which depends on the initial condition of LAQ algorithm. The constants $B_1$, $\sigma_1, \sigma_2$ and stepsize $\alpha$ should satisfy

$$\sigma_2 - (2+\sigma_2^{-\bar{t}})BMpB_1^2\tau^2 \geq \sigma_1;$$

(32a)

$$\frac{24L'^2}{\mu B_1^2} + (9p+3)\tau^2 + 9p\tau^2\sigma_2^{-\bar{t}} \leq \sigma_2;$$

(32b)

$$\alpha \geq \frac{\mu}{4L'^2M^2}.$$

(32c)

Then just by letting $P = \max\{\mathbb{V}^0, B_1^2\mathbb{V}^0\}$ we can obtain desired result (19).

In the following part, we just prove (31). First it is not difficult to verify that we can set $\mathbb{V}^0$ to be large enough to ensure (19) is satisfied for $k = -\bar{t}, -\bar{t}+1, \cdots, 0$. Then we assume that for some $k \geq 0$, (19) holds for $k - \bar{t}, k - \bar{t}+1, \cdots, k$. In the following, we need to show that (19) is true for $k+1, k+2, \cdots, k+\bar{t}+1$. It turns out that proof for $k+2, \cdots, k+\bar{t}+1$ is similar to that for $k+1$, hence, we only show the proof for $k+1$.

**1) proof of (31a) for $k+1$:**

$$\begin{aligned}\mathbb{V}(\boldsymbol{\theta}^{k+1}) &\leq \sigma_1\sigma_2^k\mathbb{V}^0 + BMpB_1^2\tau^2\sigma_2^k\mathbb{V}^0 + BMpB_1^2\tau^2(1+\sigma_2^{-T})\sigma_2^k\mathbb{V}^0\\ &= [\sigma_1 + (2+\sigma_2^{-\bar{t}})BMpB_1^2\tau^2]\sigma_2^k\mathbb{V}^0 \leq \sigma_2^{k+1}\mathbb{V}^0\end{aligned}$$

(33)

where the last inequality is the result of (32a).

**2) proof of (31b) for $k+1$:**

The following holds according to the definition of Lyapunov function:

$$f(\boldsymbol{\theta}^k) - f(\boldsymbol{\theta}^*) \leq \mathbb{V}(\boldsymbol{\theta}^k) \leq \sigma_2^k\mathbb{V}^0.$$

(34)

Assumption 1 indicates there exists a constant $L'$ such that $\|\nabla f(\boldsymbol{\theta}_1) - \nabla f(\boldsymbol{\theta}_2)\|_\infty \leq L'\|\boldsymbol{\theta}_1 - \boldsymbol{\theta}_2\|_\infty$, $\forall m \in \mathcal{M}$ and $\forall \boldsymbol{\theta}_1, \boldsymbol{\theta}_2$.

Because of convexity, the following inequality holds for any $\boldsymbol{\theta}_1$ and $\boldsymbol{\theta}_2$:

$$\langle \nabla f_m(\boldsymbol{\theta}_1) - \nabla f_m(\boldsymbol{\theta}_2), \boldsymbol{\theta}_1 - \boldsymbol{\theta}_2 \rangle \geq 0, \ \forall m \in \mathcal{M}$$

(35)

which means for any $m_1, m_2 \in \mathcal{M}$, $\nabla f_{m_1}(\boldsymbol{\theta}_1) - \nabla f_{m_1}(\boldsymbol{\theta}_2)$ and $\nabla f_{m_2}(\boldsymbol{\theta}_1) - \nabla f_{m_2}(\boldsymbol{\theta}_2)$ are of the same sign element wise. (Hint: if there exists an $i$ such that $[\nabla f_{m_1}(\boldsymbol{\theta}_1) - \nabla f_{m_1}(\boldsymbol{\theta}_2)]_i \cdot [\nabla f_{m_2}(\boldsymbol{\theta}_1) - \nabla f_{m_2}(\boldsymbol{\theta}_2)]_i < 0$, then letting all the entries other than $i$-th entry of $\boldsymbol{\theta}_1 - \boldsymbol{\theta}_2$ be zero and $[\boldsymbol{\theta}_1 - \boldsymbol{\theta}_2]_i \neq 0$ yields $\langle \nabla f_{m_1}(\boldsymbol{\theta}_1) - \nabla f_{m_1}(\boldsymbol{\theta}_2), \boldsymbol{\theta}_1 - \boldsymbol{\theta}_2 \rangle \cdot \langle \nabla f_{m_2}(\boldsymbol{\theta}_1) - \nabla f_{m_2}(\boldsymbol{\theta}_2), \boldsymbol{\theta}_1 - \boldsymbol{\theta}_2 \rangle < 0$, which contradicts (35)). Therefore, for any $\boldsymbol{\theta}_1$ and $\boldsymbol{\theta}_2$

$$\begin{aligned}\|\nabla f_m(\boldsymbol{\theta}_1) - \nabla f_m(\boldsymbol{\theta}_2)\|_\infty &\leq \|\sum_{m=1}^{M}\nabla f_m(\boldsymbol{\theta}_1) - \nabla f_m(\boldsymbol{\theta}_2)\|_\infty\\ &= \|\nabla f(\boldsymbol{\theta}_1) - \nabla f(\boldsymbol{\theta}_2)\|_\infty \leq L'\|\boldsymbol{\theta}_1 - \boldsymbol{\theta}_2\|_\infty, \ \forall m \in \mathcal{M}.\end{aligned}$$

(36)

Having this we can show

$$\|\nabla f_m(\boldsymbol{\theta}^{k+1}) - Q_m(\hat{\boldsymbol{\theta}}_m^k)\|_\infty$$

$$=\|\nabla f_m(\boldsymbol{\theta}^{k+1}) - \nabla f_m(\boldsymbol{\theta}^k) + \nabla f_m(\boldsymbol{\theta}^k) - Q_m(\boldsymbol{\theta}^k) + Q_m(\boldsymbol{\theta}^k) - Q_m(\hat{\boldsymbol{\theta}}_m^k)\|_\infty$$

$$\leq\|\nabla f_m(\boldsymbol{\theta}^{k+1}) - \nabla f_m(\boldsymbol{\theta}^k)\|_\infty + \|\nabla f_m(\boldsymbol{\theta}^k) - Q_m(\boldsymbol{\theta}^k)\|_\infty + \|Q_m(\boldsymbol{\theta}^k) - Q_m(\hat{\boldsymbol{\theta}}_m^k)\|_\infty$$

$$\leq L'\|\boldsymbol{\theta}^{k+1} - \boldsymbol{\theta}^k\|_\infty + \|\boldsymbol{\varepsilon}_m^k\|_\infty + \sqrt{\frac{1}{\alpha^2 M^2}\sum_{d=1}^D \xi_d\|\boldsymbol{\theta}^{k+1-d} - \boldsymbol{\theta}^{k-d}\|_2^2 + 3(\|\boldsymbol{\varepsilon}_m^k\|_2^2 + \|\hat{\boldsymbol{\varepsilon}}_m^{k-1}\|_2^2)}$$

$$\leq L'\sqrt{\|\boldsymbol{\theta}^{k+1} - \boldsymbol{\theta}^* + \boldsymbol{\theta}^* - \boldsymbol{\theta}^k\|_2^2} + \sqrt{\frac{1}{\alpha^2 M^2}\sum_{d=1}^D \xi_d\|\boldsymbol{\theta}^{k+1-d} - \boldsymbol{\theta}^{k-d}\|_2^2 + 3(\|\boldsymbol{\varepsilon}_m^k\|_2^2 + \|\hat{\boldsymbol{\varepsilon}}_m^{k-1}\|_2^2)} + \|\boldsymbol{\varepsilon}_m^k\|_\infty$$

$$\leq L'\sqrt{2\|\boldsymbol{\theta}^{k+1} - \boldsymbol{\theta}^*\|_2^2 + 2\|\boldsymbol{\theta}^* - \boldsymbol{\theta}^k\|_2^2} + \sqrt{\frac{1}{\alpha^2 M^2}\sum_{d=1}^D \xi_d\|\boldsymbol{\theta}^{k+1-d} - \boldsymbol{\theta}^{k-d}\|_2^2 + 3p(\|\boldsymbol{\varepsilon}_m^k\|_\infty^2 + \|\hat{\boldsymbol{\varepsilon}}_m^{k-1}\|_\infty^2)} + \|\boldsymbol{\varepsilon}_m^k\|_\infty$$

(37)

The second inequality holds because, if criterion (7) is not satisfied for $k$, $Q_m(\hat{\boldsymbol{\theta}}_m^k) - Q_m(\boldsymbol{\theta}^k) = \mathbf{0}$, otherwise, $\hat{\boldsymbol{\theta}}_m^k = \hat{\boldsymbol{\theta}}_m^{k-1}$, and $\|Q_m(\boldsymbol{\theta}^k) - Q_m(\hat{\boldsymbol{\theta}}_m^k)\|_\infty = \|Q_m(\boldsymbol{\theta}^k) - Q_m(\hat{\boldsymbol{\theta}}_m^{k-1})\|_\infty \leq \|Q_m(\boldsymbol{\theta}^k) - Q_m(\hat{\boldsymbol{\theta}}_m^{k-1})\|_2 \leq \sqrt{\frac{1}{\alpha^2 M^2}\sum_{d=1}^D \xi_d\|\boldsymbol{\theta}^{k+1-d} - \boldsymbol{\theta}^{k-d}\|_2^2 + 3(\|\boldsymbol{\varepsilon}_m^k\|_2^2 + \|\hat{\boldsymbol{\varepsilon}}_m^{k-1}\|_2^2)}$.

Because of Assumption 2, the following holds

$$\|\boldsymbol{\theta} - \boldsymbol{\theta}^*\|_2^2 \leq \frac{2}{\mu}[f(\boldsymbol{\theta}) - f(\boldsymbol{\theta}^*)].$$

(38)

Thus, we have

$$\|\nabla f_m(\boldsymbol{\theta}^{k+1}) - Q_m(\hat{\boldsymbol{\theta}}_m^k)\|_\infty^2$$

$$\leq 3\{\frac{4L'^2}{\mu}[f(\boldsymbol{\theta}^{k+1}) - f(\boldsymbol{\theta}^*) + f(\boldsymbol{\theta}^k) - f(\boldsymbol{\theta}^*)] + \frac{1}{\alpha^2 M^2}\sum_{d=1}^D \xi_d\|\boldsymbol{\theta}^{k+1-d} - \boldsymbol{\theta}^{k-d}\|_2^2\}$$

$$+ 9p(\|\boldsymbol{\varepsilon}_m^k\|_\infty^2 + \|\hat{\boldsymbol{\varepsilon}}_m^{k-1}\|_\infty^2) + 3\|\boldsymbol{\varepsilon}_m^k\|_\infty^2$$

(39)

$$\leq \frac{12L'^2}{\mu}[f(\boldsymbol{\theta}^{k+1}) - f(\boldsymbol{\theta}^*) + f(\boldsymbol{\theta}^k) - f(\boldsymbol{\theta}^*) + \frac{\mu}{4L'^2\alpha^2 M^2}\sum_{d=1}^D \xi_d\|\boldsymbol{\theta}^{k+1-d} - \boldsymbol{\theta}^{k-d}\|_2^2]$$

$$+ (9p + 3)\|\boldsymbol{\varepsilon}_m^k\|_\infty^2 + 9p\|\hat{\boldsymbol{\varepsilon}}_m^{k-1}\|_\infty^2.$$

Since (32c) holds,

$$\frac{\mu\xi_d}{4L'^2\alpha^2 M^2} \leq \frac{\xi_d}{\alpha} \leq \sum_{j=d}^D \frac{\xi_d}{\alpha}, \quad \forall d \in \{1, 2, \cdots, D\}.$$

(40)

Substituting (40) into (39) yields:

$$\|\nabla f_m(\boldsymbol{\theta}^{k+1}) - Q_m(\hat{\boldsymbol{\theta}}_m^k)\|_\infty^2$$

$$\leq \frac{12L'^2}{\mu}[f(\boldsymbol{\theta}^{k+1}) - f(\boldsymbol{\theta}^*) + f(\boldsymbol{\theta}^k) - f(\boldsymbol{\theta}^*) + \sum_{d=1}^D\sum_{j=d}^D \frac{\xi_j}{\alpha}\|\boldsymbol{\theta}^{k+1-d} - \boldsymbol{\theta}^{k-d}\|_2^2] + (9p+3)\|\boldsymbol{\varepsilon}_m^k\|_\infty^2$$

$$+ 9p\|\hat{\boldsymbol{\varepsilon}}_m^{k-1}\|_\infty^2$$

(41)

$$\leq \frac{12L'^2}{\mu}[\mathbb{V}(\boldsymbol{\theta}^{k+1}) + \mathbb{V}(\boldsymbol{\theta}^k)] + (9p+3)\|\boldsymbol{\varepsilon}_m^k\|_\infty^2 + 9p\|\hat{\boldsymbol{\varepsilon}}_m^{k-1}\|_\infty^2$$

$$\leq \frac{24L'^2}{\mu}\sigma_2^k\mathbb{V}^0 + (9p+3)B_1\tau\sigma_2^k\mathbb{V}^0 + 9pB_1\tau\sigma_2^{(k-\bar{t})}\mathbb{V}^0$$

$$= [\frac{24L'^2}{\mu} + (9p+3)B_1^2\tau^2 + 9pB_1^2\tau^2\sigma_2^{-\bar{t}}]\sigma_2^k\mathbb{V}^0 \leq B_1^2\sigma_2^{(k+1)}\mathbb{V}^0$$

where the second inequality follows from (34) and the last equality is the result of (32b). Therefore,

$$\|\boldsymbol{\varepsilon}_m^{k+1}\|_\infty^2 \leq \tau^2(R_m^k)^2 = \tau^2\|\nabla f_m(\boldsymbol{\theta}^{k+1}) - Q_m(\hat{\boldsymbol{\theta}}_m^k)\|_\infty^2 \leq B_1^2\tau^2\sigma_2^{k+1}\mathbb{V}^0.$$

(42)

Here we have finished the proof that (31) hold for any integer $k \geq 0$.

Now we show that there do exist $\sigma_1 \in (0,1)$ and $\sigma_2 \in (0,1)$ such that (17) and (32) are satisfied. First, we fix $\rho_1 = \frac{1}{2}$, $\rho_2 = 1$, and $\xi_1 = \xi_2 = \cdots = \xi_D = \xi$, which reduces (17) as:

$$D\xi \leq \frac{1}{16}; \tag{43a}$$

$$\alpha \leq \frac{1}{L}\left(\frac{1}{8} - 2D\xi\right). \tag{43b}$$

Thus, we set $D\xi = \frac{1}{32}$ and $\alpha = \frac{\eta}{32L}$, $\eta \in (0,1)$. With the condition number $\kappa := \frac{L}{\mu} \geq 1$, it follows that

$$c = \min\{\frac{\eta(2-\eta)}{256\kappa}, \frac{14-\eta}{32}, \frac{14-\eta}{32(D-d+1)}\Big|_{d=1}^{D-1}\} = \min\{\frac{\eta(2-\eta)}{256\kappa}, \frac{14-\eta}{32D}\}; \tag{44a}$$

$$B = \frac{3\eta(\eta+18)M}{1024L}. \tag{44b}$$

We choose $D \leq \kappa$, the it can be verify $\frac{\eta(2-\eta)}{256\kappa} < \frac{14-\eta}{32D}$ holds. Hence,

$$\sigma_1 = 1 - c = 1 - \frac{\eta(2-\eta)}{256\kappa}. \tag{45}$$

Above values enforce (17) satisfied. Then we check (32). That (32c) holds means

$$\eta \geq \frac{8\mu L}{L'^2 M^2} \approx \frac{8}{\kappa M^2} \tag{46}$$

where we use $L \approx L'$. In practical problem, $\frac{8}{\kappa M^2} < 1$ usually holds. Then we first show a necessary condition for (32a):

$$1 - B \geq \sigma_1 \tag{47}$$

which with $B$ and $\sigma_1$ in (44b) and (45) plugged in is equivalent to the following :

$$\eta < \frac{8\mu - 54M}{4\mu + 3M}, \quad \text{with} \quad \mu > \frac{27M}{4}. \tag{48}$$

Note that $\mu > 27M/4$ can be achieved by scaling the loss function. Scaling the loss function does not change the learning problem and does not changes the condition constant $\kappa$. It worths mentioning that it is only when $\rho_1 = \frac{1}{2}$ and $\rho_2 = 1$ (46) and (48) should hold. Actually, $\rho_1$ and $\rho_2$ are only constrained as $0 < \rho_1 < 1$ and $\rho_2 > 0$. Consequently, $\eta$ has a lager range of choice instead of only in the range described by (46) and (48).

For any $\sigma_2 \in (0,1)$, there exists a $\bar{t} \geq 1$ such that $\sigma_2^{-\bar{t}} \leq B_2$, where $B_2$ is not too large. Define $\eta' := B_1^2 \tau^2$, then a sufficient condition of (32a) and (32b) is

$$\sigma_1 + (2 + B_2)BM^2 p\eta') \leq \sigma_2 < 1; \tag{49a}$$

$$[\frac{24L'^2}{\mu\eta'} + 9p(B_2+1) + 3]\tau^2 \leq \sigma_2. \tag{49b}$$

When $\eta'$ is chosen to be small enough and $\sigma_2$ is close enough to 1, (49a) is equivalent to (47). Hence, choosing $\eta$ satisfying (46) and (48) is sufficient to guarantee (49a). With $\eta'$ fixed, we can let $\tau$ to be small enough to ensure that (49b) holds. So far, we have shown that we can find $\sigma_1$ and $\sigma_2$ satisfying $0 < \sigma_1 < \sigma_2 < 1$, thus validate LAQ converges at linear rate.

# D   Alternative proof of Theorem 1 based on a new Lyapunov function

For this proof we define Lyapunov function as

$$\mathbb{V}(\boldsymbol{\theta}^k) := f(\boldsymbol{\theta}^k) - f(\boldsymbol{\theta}^*) + \sum_{d=1}^{D}\sum_{j=d}^{D}\frac{\xi_j}{\alpha}\|\boldsymbol{\theta}^{k+1-d} - \boldsymbol{\theta}^{k-d}\|_2^2 + \gamma\sum_{m\in\mathcal{M}}\|\boldsymbol{\varepsilon}_m^k\|_\infty^2 \tag{50}$$

which differentiates from the that defined in the paper in that the error is included.

$$\|\boldsymbol{\varepsilon}_m^{k+1}\|_\infty^2 \leq \tau^2(R_m^{k+1})^2 = \tau^2\|\nabla f_m(\boldsymbol{\theta}^{k+1}) - \nabla f_m(\boldsymbol{\theta}^k) + \nabla f_m(\boldsymbol{\theta}^k) - Q_m(\boldsymbol{\theta}^k) + Q_m(\boldsymbol{\theta}^k) - Q_m(\hat{\boldsymbol{\theta}}_m^k)\|_\infty^2$$

$$\leq 3\tau^2 L'\|\boldsymbol{\theta}^{k+1} - \boldsymbol{\theta}^k\|_2^2 + 3\tau^2\|\boldsymbol{\varepsilon}_m^k\|_\infty^2 + 3\tau^2\|Q_m(\boldsymbol{\theta}^k) - Q_m(\hat{\boldsymbol{\theta}}_m^k)\|_2^2 \tag{51}$$

Then the one-step Lyapunov function difference is bounded as

$$
\mathbb{V}(\boldsymbol{\theta}^{k+1}) - \mathbb{V}(\boldsymbol{\theta}^k)
$$

$$
\leq -\alpha \left\langle \nabla f(\boldsymbol{\theta}^k), Q(\boldsymbol{\theta}^k) \right\rangle + \frac{\alpha}{2}\|\nabla f(\boldsymbol{\theta}^k)\|_2^2 + \frac{\alpha}{2}\| \sum_{m \in \mathcal{M}_c^k} Q_m(\hat{\boldsymbol{\theta}}_m^{k-1}) - Q_m(\boldsymbol{\theta}^k)\|_2^2
$$

$$
+ (\frac{L}{2} + \beta_1 + 3\gamma\tau^2 L'^2)\|\boldsymbol{\theta}^{k+1} - \boldsymbol{\theta}^k\|_2^2 + \sum_{d=1}^{D-1}(\beta_{d+1} - \beta_d)\|\boldsymbol{\theta}^{k+1-d} - \boldsymbol{\theta}^{k-d}\|_2^2 - \beta_D\|\boldsymbol{\theta}^{k+1-D} - \boldsymbol{\theta}^{k-D}\|_2^2
$$

$$
+ \gamma(3\tau^2 - 1) \sum_{m \in \mathcal{M}} \|\varepsilon_m^k\|_\infty^2 + 3\gamma\tau^2 \sum_{m \in \mathcal{M}} \|Q_m(\hat{\boldsymbol{\theta}}_m^{k-1}) - Q_m(\boldsymbol{\theta}^k)\|_2^2
$$

$$
\leq -\alpha \left\langle \nabla f(\boldsymbol{\theta}^k), Q(\boldsymbol{\theta}^k) \right\rangle + \frac{\alpha}{2}\|\nabla f(\boldsymbol{\theta}^k)\|_2^2 + [\frac{\alpha}{2} + (\frac{L}{2} + \beta_1 + 3\gamma\tau^2 L'^2)(1 + \rho_2^{-1})\alpha^2]\| \sum_{m \in \mathcal{M}_c^k} Q_m(\hat{\boldsymbol{\theta}}_m^{k-1}) - Q_m(\boldsymbol{\theta}^k)\|_2^2
$$

$$
+ (\frac{L}{2} + \beta_1 + 3\gamma\tau^2 L'^2)(1 + \rho_2)\alpha^2\|Q(\boldsymbol{\theta}^k)\|_2^2 + \sum_{d=1}^{D-1}(\beta_{d+1} - \beta_d)\|\boldsymbol{\theta}^{k+1-d} - \boldsymbol{\theta}^{k-d}\|_2^2
$$

$$
- \beta_D\|\boldsymbol{\theta}^{k+1-D} - \boldsymbol{\theta}^{k-D}\|_2^2 + \gamma(3\tau^2 - 1) \sum_{m \in \mathcal{M}} \|\varepsilon_m^k\|_\infty^2 + 3\gamma\tau^2 \sum_{m \in \mathcal{M}} \|Q_m(\hat{\boldsymbol{\theta}}_m^{k-1}) - Q_m(\boldsymbol{\theta}^k)\|_2^2
$$

$$
\leq -\frac{\alpha}{2}\|\nabla f(\boldsymbol{\theta}^k)\|_2^2 + \alpha \left\langle \nabla f(\boldsymbol{\theta}^k), \varepsilon^k \right\rangle + [(\frac{\alpha}{2} + (\frac{L}{2} + \beta_1)(1 + \rho_2^{-1})\alpha^2)M + 3\gamma\tau^2]\frac{1}{\alpha^2 M} \sum_{d=1}^{D} \xi_d\|\boldsymbol{\theta}^{k+1-d} - \boldsymbol{\theta}^{k-d}\|_2^2
$$

$$
+ (\frac{L}{2} + \beta_1 + 3\gamma\tau^2 L'^2)(1 + \rho_2)\alpha^2\|\nabla f(\boldsymbol{\theta}^k) - \varepsilon^k\|_2^2 + \sum_{d=1}^{D-1}(\beta_{d+1} - \beta_d)\|\boldsymbol{\theta}^{k+1-d} - \boldsymbol{\theta}^{k-d}\|_2^2 - \beta_D\|\boldsymbol{\theta}^{k+1-D} - \boldsymbol{\theta}^{k-D}\|_2^2
$$

$$
+ [\frac{3\alpha}{2} + (\frac{3L}{2} + 3\beta_1 + 9\gamma\tau^2 L'^2)(1 + \rho_2^{-1})\alpha^2 + 3\gamma\tau^2]M \sum_{m \in \mathcal{M}_c^k} (\|\varepsilon_m^k\|_2^2 + \|\hat{\varepsilon}_m^{k-1}\|_2^2) + \gamma(3\tau^2 - 1) \sum_{m \in \mathcal{M}} \|\varepsilon_m^k\|_\infty^2
$$

$$
\leq [(-\frac{1}{2} + \frac{1}{2}\rho_1)\alpha + (L + 2\beta_1 + 6\gamma\tau^2 L'^2)(1 + \rho_2)\alpha^2]\|\nabla f(\boldsymbol{\theta}^k)\|_2^2
$$

$$
+ \{[(\frac{\alpha}{2} + (\frac{L}{2} + \beta_1 + 3\gamma\tau^2 L'^2)(1 + \rho_2^{-1})\alpha^2)M + 3\gamma\tau^2]\frac{\xi_D}{\alpha^2 M} - \beta_D\}\|\boldsymbol{\theta}^{k+1-D} - \boldsymbol{\theta}^{k-D}\|_2^2
$$

$$
+ \sum_{d=1}^{D-1}\{[(\frac{\alpha}{2} + (\frac{L}{2} + \beta_1 + 3\gamma\tau^2 L'^2)(1 + \rho_2^{-1})\alpha^2)M + 3\gamma\tau^2]\frac{\xi_d}{\alpha^2 M} + \beta_{d+1} - \beta_d\}\|\boldsymbol{\theta}^{k+1-d} - \boldsymbol{\theta}^{k-d}\|_2^2
$$

$$
+ [\frac{3\alpha}{2} + (\frac{3L}{2} + 3\beta_1 + 9\gamma\tau^2 L'^2)(1 + \rho_2^{-1})\alpha^2 + 3\gamma\tau^2]M \sum_{m \in \mathcal{M}_c^k} (\|\varepsilon_m^k\|_2^2 + \|\hat{\varepsilon}_m^{k-1}\|_2^2)
$$

$$
+ [\frac{1}{2\rho_1}\alpha + (L + 2\beta_1 + 6\gamma\tau^2 L'^2)(1 + \rho_2)\alpha^2]\|\varepsilon^k\|_2^2 + \gamma(3\tau^2 - 1) \sum_{m \in \mathcal{M}} \|\varepsilon_m^k\|_\infty^2
$$

$$
\tag{52}
$$

where the second inequality uses the Young' inequality and the third inequality follows from (23).

It is straightforward that the following condition guarantees the first three terms in above inequality are nonpositive

$$
(-\frac{1}{2} + \frac{1}{2}\rho_1)\alpha + (L + 2\beta_1 + 6\gamma\tau^2 L'^2)(1 + \rho_2)\alpha^2 \leq 0;
$$

$$
[(\frac{\alpha}{2} + (\frac{L}{2} + \beta_1 + 3\gamma\tau^2 L'^2)(1 + \rho_2^{-1})\alpha^2)M + 3\gamma\tau^2]\frac{\xi_D}{\alpha^2 M} - \beta_D \leq 0; \tag{53}
$$

$$
[(\frac{\alpha}{2} + (\frac{L}{2} + \beta_1 + 3\gamma\tau^2 L'^2)(1 + \rho_2^{-1})\alpha^2)M + 3\gamma\tau^2]\frac{\xi_d}{\alpha^2 M} + \beta_{d+1} - \beta_d \leq 0.
$$

For the ease of exposition, we define constant $c$ and $B$ as

$$
c = \min\{(1 - \rho_1)\alpha - 2\mu(L + 2\beta_1 + 6\gamma\tau^2 L'^2)(1 + \rho_2)\alpha^2,
$$

$$
1 - [(\frac{\alpha}{2} + (\frac{L}{2} + \beta_1 + 3\gamma\tau^2 L'^2)(1 + \rho_2^{-1})\alpha^2)M + 3\gamma\tau^2]\frac{\xi_D}{\alpha^2 M \beta_D}, \tag{54}
$$

$$
1 - \frac{\beta_{d+1}}{\beta_d} - [(\frac{\alpha}{2} + (\frac{L}{2} + \beta_1 + 3\gamma\tau^2 L'^2)(1 + \rho_2^{-1})\alpha^2)M + 3\gamma\tau^2]\frac{\xi_d}{\alpha^2 M \beta_d}\}
$$

and,

$$B = \max\{[\frac{3\alpha}{2}+(\frac{3L}{2}+3\beta_1+9\gamma\tau^2 L'^2)(1+\rho_2^{-1})\alpha^2+3\gamma\tau^2]M, \frac{1}{2\rho_1}\alpha+(L+2\beta_1+6\gamma\tau^2 L'^2)(1+\rho_2)\alpha^2\}.$$
(55)

Assumption 2 indicates $f(\cdot)$ satisfies the PL condition:

$$2\mu(f(\boldsymbol{\theta}^k)-f(\boldsymbol{\theta}^*)) \leq \|\nabla f(\boldsymbol{\theta}^k)\|_2^2.$$
(56)

Plugging (56) into (52) gives

$$\mathbb{V}(\boldsymbol{\theta}^{k+1}) \leq \sigma_1 \mathbb{V}(\boldsymbol{\theta}^k) + B[\|\varepsilon^k\|_2^2 + \sum_{m\in\mathcal{M}} (\|\varepsilon_m^k\|_2^2 + \|\hat{\varepsilon}_m^{k-1}\|_2^2)] + \gamma(3\tau^2-1)\sum_{m\in\mathcal{M}}\|\varepsilon_m^k\|_\infty^2$$
$$\leq \sigma_1 \mathbb{V}(\boldsymbol{\theta}^k) + [BMp^2 + B + \gamma(3\tau^2-1)]\sum_{m\in\mathcal{M}}\|\varepsilon_m^k\|_2^2 + Bp^2\sum_{m\in\mathcal{M}}\|\hat{\varepsilon}_m^{k-1}\|_2^2$$
(57)

where $\sigma_1 = 1-c$.

By choosing parameter stepsize $\alpha$ that impose the following inequality hold

$$[BMp^2 + B + \gamma(3\tau^2-1)] \leq 0,$$
(58)

one can obtain

$$\mathbb{V}(\boldsymbol{\theta}^{k+1}) \leq \sigma_1 \mathbb{V}(\boldsymbol{\theta}^k) + Bp^2\frac{1}{\gamma}\cdot\gamma\sum_{m\in\mathcal{M}}\|\hat{\varepsilon}_m^{k-1}\|_2^2$$
$$\leq \sigma_1 \mathbb{V}(\boldsymbol{\theta}^k) + Bp^2\frac{1}{\gamma}\sum_{m\in\mathcal{M}}\max_{k-\bar{t}\leq t'\leq k-1}\mathbb{V}(\boldsymbol{\theta}^{t'})$$
$$\leq \sigma_1 \mathbb{V}(\boldsymbol{\theta}^k) + BMp^2\frac{1}{\gamma}\max_{k-\bar{t}\leq t'\leq k-1}\mathbb{V}(\boldsymbol{\theta}^{t'}).$$
(59)

For simplicity, we fix $\rho_1 = \frac{1}{2}$, $\rho_2 = 1$, $\beta_d = \frac{(D-d+1)\xi}{\alpha}$, $\alpha = \frac{a}{L}$, and $\gamma\tau^2 = \frac{bL}{L'^2}$, with $a, b > 0$. Consequently, we obtain

$$B = [\frac{3\alpha}{2}+(\frac{3L}{2}+3\beta_1+9\gamma\tau^2 L'^2)\alpha^2+3\gamma\tau^2]M$$
$$= [\frac{3a}{2L}+(\frac{3a}{2}+3D\xi+9ab)\frac{2a}{L}+\frac{9bL}{ML'^2}]M$$
(60)

and

$$c = \min\left\{\frac{[\frac{1}{2}-4(a+2D\xi+6ab)]a}{\kappa}, \frac{\frac{1}{2}-(\frac{1}{2}a+D\xi+3ab)+\frac{3bL^2}{aL'^2 M}}{D-d+1}\right\}.$$
(61)

For the design parameter $D$, we impose $D \leq \kappa$. From (61), it is obvious that the following condition

$$[\frac{1}{2}-4(a+2D\xi+6ab)]a \leq \frac{1}{2}-(\frac{1}{2}a+D\xi+3ab)+\frac{3bL^2}{aL'^2 M}$$
(62)

guarantees

$$c = \frac{[\frac{1}{2}-4(a+2D\xi+6ab)]a}{\kappa}.$$
(63)

Thus, we obtain $\sigma_1 = 1-c = 1-\frac{[\frac{1}{2}-4(a+2D\xi+6ab)]a}{\kappa}$.

Following [8, Lemma 3.2], if the following condition holds

$$\sigma_1 + BMp^2\frac{1}{\gamma} < 1$$
(64)

then it guarantees the linear convergence of $\mathbb{V}$, that is,

$$\mathbb{V}(\boldsymbol{\theta}^k) \leq \sigma_2^k \mathbb{V}(\boldsymbol{\theta}^k)$$
(65)

where $\sigma_2 = (\sigma_1 + BMp^2\frac{1}{\gamma})^{\frac{1}{1+\bar{t}}}$. It can be verified that $a = \frac{1}{20}$, $b = \frac{1}{10}$, $D\xi = \frac{1}{50}$ and $\tau^2 \leq \frac{1}{100\kappa}/[M^2 p^2(\frac{93L'^2}{10L^2}+\frac{9}{M})]$ is a sufficient condition for (53), (62) and (64) being satisfied. Therefore, the linear convergence of (65) is indeed guaranteed. With above selected parameters, we can obtain $\sigma_1 = 1-\frac{1}{1000\kappa}$ and $\sigma_2 = (1-\frac{1}{1000\kappa}+M^2 p^2(\frac{93L'^2}{100L^2}+\frac{9}{10M})\tau^2)^{\frac{1}{1+\bar{t}}}$. It is thus obvious that with the quantization being accurate enough, i.e., $\tau^2 \to 0$, the dependence of convergence rate on condition number is of order $\frac{1}{\kappa}$, which is the same as standard gradient descent.

# E  Proof of Proposition [1]

Suppose that at current iteration $k$ the last iteration when worker $m$ communicated with server is $d'$ where $1 \le d' \le d_m$, then $\boldsymbol{\theta}_m^{k-1} = \boldsymbol{\theta}^{k-d'}$. Therefore,

$$
\begin{aligned}
\|Q_m(\hat{\boldsymbol{\theta}}_m^{k-1}) - Q_m(\boldsymbol{\theta}^k)\|_2^2 =& \|Q_m(\boldsymbol{\theta}^{k-d'}) - \nabla f_m(\boldsymbol{\theta}^{k-d'}) - Q_m(\boldsymbol{\theta}^k) + \nabla f_m(\boldsymbol{\theta}^k) + \nabla f_m(\boldsymbol{\theta}^{k-d'}) - \nabla f_m(\boldsymbol{\theta}^k)\|_2^2 \\
\le& 3(\|f_m(\boldsymbol{\theta}^{k-d'}) - \nabla f_m(\boldsymbol{\theta}^k)\|_2^2 + \|\boldsymbol{\varepsilon}_m^k\|_2^2 + \|\boldsymbol{\varepsilon}_m^{k-d'}\|_2^2) \\
\le& 3L_m^2 \|\boldsymbol{\theta}^{k-d'} - \boldsymbol{\theta}^k\|_2^2 + 3(\|\boldsymbol{\varepsilon}_m^k\|_2^2 + \|\boldsymbol{\varepsilon}_m^{k-d'}\|_2^2) \\
=& 3L_m^2 \|\sum_{d=1}^{d'} \boldsymbol{\theta}^{k+1-d} - \boldsymbol{\theta}^{k-d}\|_2^2 + 3(\|\boldsymbol{\varepsilon}_m^k\|_2^2 + \|\boldsymbol{\varepsilon}_m^{k-d'}\|_2^2) \\
\le& 3L_m^2 d' \sum_{t=1}^{d'} \|\boldsymbol{\theta}^{k+1-d} - \boldsymbol{\theta}^{k-d}\|_2^2 + 3(\|\boldsymbol{\varepsilon}_m^k\|_2^2 + \|\boldsymbol{\varepsilon}_m^{k-d'}\|_2^2).
\end{aligned}
\tag{66}
$$

From the definition of $d_m$ and $\xi_1 \ge \xi_2 \ge \cdots \ge \xi_D$, it can be obtained that:

$$
L_m^2 \le \frac{\xi_{d'}}{3\alpha^2 M^2 D}, \text{ for all } d' \text{ satisfying } 1 \le d' \le d_m.
\tag{67}
$$

Substituting (67) into (66) gives:

$$
\begin{aligned}
\|Q_m(\hat{\boldsymbol{\theta}}_m^{k-1}) - Q_m(\boldsymbol{\theta}^k)\|_2^2 \le& \frac{\xi_{d'}}{\alpha^2 M^2} \sum_{d=1}^{d'} \xi_d \|\boldsymbol{\theta}^{k+1-d} - \boldsymbol{\theta}^{k-d}\|_2^2 + 3(\|\boldsymbol{\varepsilon}_m^k\|_2^2 + \|\hat{\boldsymbol{\varepsilon}}_m^{k-1}\|_2^2) \\
\le& \frac{1}{\alpha^2 M^2} \sum_{d=1}^{D} \xi_d \|\boldsymbol{\theta}^{k+1-d} - \boldsymbol{\theta}^{k-d}\|_2^2 + 3(\|\boldsymbol{\varepsilon}_m^k\|_2^2 + \|\hat{\boldsymbol{\varepsilon}}_m^{k-1}\|_2^2)
\end{aligned}
\tag{68}
$$

which exactly means that (7a) is satisfied. Since $d_m \le D \le \bar{t}$, the criterion (7) holds, which means that worker $m$ will not upload its information until at least $t_m$ iterations after last communication. therefore, in first $k$ iterations, worker $m$ has at most $k/(d_m + 1)$ communications with the server.

# F  Intuition of the selective aggregation criterion (7a)

The following part shows the inspiration for the criterion, which is not mathematically strict but provides the intuition. For simplicity, we fix $\alpha = 1/L$, then we have:

$$
\begin{aligned}
\Delta_{GD}^k &= -\frac{\alpha}{2}\|\nabla f(\boldsymbol{\theta}^k)\|_2^2; \\
\Delta_{LAQ}^k &= -\frac{\alpha}{2}\|\nabla f(\boldsymbol{\theta}^k)\|_2^2 - \alpha\|\sum_{m \in \mathcal{M}_c^k}(Q_m(\hat{\boldsymbol{\theta}}_m^{k-1}) - Q_m(\boldsymbol{\theta}^k))\|_2^2.
\end{aligned}
\tag{69}
$$

The lazy aggregation criterion selects the quantized gradient innovation by judging its contribution to decreasing the loss function. LAQ is expected to be more communication-efficient than GD, that is, each upload results more descent, which translates to:

$$
\frac{\Delta_{LAQ}^k}{|\mathcal{M}^k|} \le \frac{\Delta_{GD}^k}{M}
\tag{70}
$$

By simple manipulation, it can be obtained that (70) is equivalent to:

$$
\|\sum_{m \in \mathcal{M}_c^k}(Q_m(\hat{\boldsymbol{\theta}}_m^{k-1}) - Q_m(\boldsymbol{\theta}^k))\|_2^2 \le \frac{|\mathcal{M}_c^k|}{2M}\|\nabla f(\boldsymbol{\theta}^k)\|_2^2.
\tag{71}
$$

Since

$$
\|\sum_{m \in \mathcal{M}_c^k}(Q_m(\hat{\boldsymbol{\theta}}_m^{k-1}) - Q_m(\boldsymbol{\theta}^k))\|_2^2 \le |\mathcal{M}_c^k| \sum_{m \in \mathcal{M}_c^k}\|(Q_m(\hat{\boldsymbol{\theta}}_m^{k-1}) - Q_m(\boldsymbol{\theta}^k)\|_2^2,
\tag{72}
$$

the following condition is sufficient to guarantee (71):

$$
\|(Q_m(\hat{\boldsymbol{\theta}}_m^{k-1}) - Q_m(\boldsymbol{\theta}^k)\|_2^2 \le \|\nabla f(\boldsymbol{\theta}^k)\|_2^2/(2M^2), \ \forall m \in \mathcal{M}_c^k.
\tag{73}
$$

However, to check (73) locally for each worker is impossible because the fully aggregated gradient $\nabla f(\boldsymbol{\theta}^k)$ is required, which is exactly what we want to avoid. Moreover, it does not make sense to reduce uploads if the fully aggregated gradient has been obtained. Therefore, we bypass directly calculating $||\nabla f(\boldsymbol{\theta}^k)||_2^2$ using its approximation below.

$$||\nabla f(\boldsymbol{\theta}^k)||_2^2 \approx \frac{2}{\alpha^2} \sum_{k=1}^{D} \xi_d ||\boldsymbol{\theta}^{k+1-d} - \boldsymbol{\theta}^{k-d}||_2^2 \tag{74}$$

where $\{\xi_d\}_{d=1}^{D}$ are constants. The fundamental reason why (74) holds is that $\nabla f(\boldsymbol{\theta}^k)$ can be approximated by weighted previous gradients or parameter differences since $f(\cdot)$ is $L$-smooth. Combining (74) and (73) leads to proposed criterion (7a) with quantization error ignored.

## G   Simulation details

**Logistic regression** In multi-class logistic regression, suppose there are $C$ classes, for instance, in MNIST dataset, $C = 10$. The training data $\mathbf{x}_{m,n}$ is denoted as feature-label pair $(\mathbf{x}_{m,n}^f, \mathbf{x}_{m,n}^l)$, where $\mathbf{x}_{m,n}^f \in \mathbb{R}^F$ is the feature vector and $\mathbf{x}_{m,n}^l \in \mathbb{R}^C$ is the one-hot label vector. Hence the model parameter $\boldsymbol{\theta} \in R^{C \times F}$ is a matrix, which is slightly different from previous description. Note that the model is formulated in this way for the convenience of expression, which does not change the learning problem. The estimated probability of $(m, n)$-th sample belonging to class $i$ is given by

$$\hat{\mathbf{x}}_{m,n}^l = \text{softmax}(\boldsymbol{\theta}\mathbf{x}_{m,n}^f) \tag{75}$$

which can be explicitly written as:

$$[\hat{\mathbf{x}}_{m,n}^l]_i = \frac{e^{[\boldsymbol{\theta}\mathbf{x}_{m,n}^f]_i}}{\sum_{j=1}^{C} e^{[\boldsymbol{\theta}\mathbf{x}_{m,n}^f]_j}}, \, \forall i \in \{1, 2, \cdots, C\}. \tag{76}$$

Regularized logistic regression adopts loss as cross-entropy plus regularizer:

$$\ell(\mathbf{x}_{m,n}, \boldsymbol{\theta}) = -\sum_{i}^{C} [\mathbf{x}_{m,n}^l]_i \log[\hat{\mathbf{x}}_{m,n}^l]_i + \frac{\lambda}{2} Tr(\boldsymbol{\theta}^T \boldsymbol{\theta}) \tag{77}$$

where $Tr(\cdot)$ denotes trace operator, and $\boldsymbol{\theta}^T$ is the transpose of $\boldsymbol{\theta}$. With $\ell(\mathbf{x}_{m,n}, \boldsymbol{\theta})$ defined, the local loss functions can be determined as $f_m(\boldsymbol{\theta}) = \sum_{n=1}^{N_m} \ell(\mathbf{x}_{m,n}; \boldsymbol{\theta})$, and the global loss function adopts following form:

$$f(\boldsymbol{\theta}) = \frac{1}{N} \sum_{m \in \mathcal{M}} f_m(\boldsymbol{\theta}) \tag{78}$$

where $N$ is the total number of data samples. In our tests, the regularizer coefficient $\lambda$ is $0.01$.

**Neural network.** We employ a ReLU network of one hidden layer with $200$ nodes, the dimensions of input layer and output layer are $784$ and $10$, respectively. The regularizer parameter $\lambda = 0.01$.