[Reviews · NeurIPS 2019]

Reviewer 1



============= After Author Response ==================== I have read the authors' rebuttal, my evaluation remains unchanged. =================================================== This paper proposed lazily aggregated quantized gradient methods for communication-efficient distributed learning. The main idea is to combine lazily aggregated gradient and quantized gradient into a single framework to further save communication cost in distributed learning problems. The proposed method looks reasonable, and convergence analysis is provided showing that linear convergence is achieved under certain conditions for strongly convex and smooth functions. Empirical results were provided showing that the proposed approach improves both quantized gradient methods or lazily aggregated methods. My main concern on the theoretical analysis, which is done only on strongly convex and smooth functions, also the dependencies on the condition numbers (smoothness and strongly convex) are hidden in the main paper. By looking at the supplemental materials, it appears that the condition number dependency are quite bad and being much worse than standard gradient methods.

Reviewer 2



The paper extends the lazily aggregated gradient (LAG) approach by applying quantization to further reduce communication. In the original LAG approach, workers only communicate their gradient with the central coordinator if it is significantly different from its previous one. In this paper, the gradients are compressed using quantization and workers skip communication if their quantized gradient does not differ substantially from previous ones. For strongly convex objectives, the paper proves linear convergence. The paper is very well written and the approach is clearly motivated, easy to understand, and discussed in the context of related work. Although the proposed approach is a straight-forward extension of the LAG approach, the idea is sound and theoretically evaluated. The theoretical analysis is performed using Lyapunov functions that measure not only the loss difference between an iterate and the optimal model, but also include past gradient differences and the quantization error. The relation of this measure to the simple (expected) risk should be discussed in more detail. Also, this analysis does not apply to neural networks (for the proofs, the authors assume strong convexity). The empirical analysis shows a substantial communication reduction over gradient descent, quantized gradient descent and LAG, while retaining the same test accuracy. My only concern is that the experiments have been only conducted on MNIST. There, comparable accuracies can be reached easily (e.g., by averaging aggregates only once at the end, as in [1]). Thus, it is unclear how the approach performs in practice. Overall the paper is a nice extension to the LAG approach with a solid theoretical analysis. Despite my concerns about the empirical evaluation, I think the paper is a nice contribution to the conference. Detailed comments: - the approach is motivated by the need to reduce the number of worker-to-server uplink communication, because of the latency through sequential uploads. However, in [4] it was shown that asynchronous updates work in practice. It would be interesting to discuss this approach with respect to quantization, since this might lead to more collisions and thus render [4] infeasible. - it would be very interesting to see how LAQ compares to randomly selecting workers that upload (as in [2]). In this light, it would be also interesting to discuss the advantages of LAQ over approaches that dynamically silence all workers (as in [3]). - the proof of theorem 1 was quite difficult to understand and the Lyapunov-functions are not very intuitive to me. Thus it would be great if the authors could add more intuitive explanations to the proof in the supplementary material. - the authors claim in the reproducibility checklist that their code is available but I couldn't find a link to it in the paper. - typo in line 66: "see also" [1] Zinkevich, et al., "Parallel Stochastic Gradient Descent", NIPS 2010 [2] McMahan et al. "Communication-Efficient Learning of Deep Networks from Decentralized Data", AISTATS 2017 [3] Kamp, et al. "Efficient Decentralized Deep Learning by Dynamic Model Averaging", ECMLPKDD 2018 [4] Recht, et al., "Hogwild: A lock-free approach to parallelizing stochastic gradient descent", NIPS 2011 The authors responded to all my comments and I remain convinced that this paper would be a good contribution to the conference.

Reviewer 3



The authors consider the problem of reducing the amount of communication for learning in a parameter-server setting. The approach is based on a combination of quantization of new gradient information and skipping updates when innovations are not considered informative enough. In particular, it combines the idea of lazily aggregating gradients (NeurIPS 2018) with quantization of the gradient components.  The combination of quantization of elements and skipping updates is interesting and potentially useful. However, both the paper and the scheme itself has some limitations which would be nice to have addressed. 1. The scheme requires additional memory, both at the server (to compute update direction in absence of new information) and at the workers (e.g. to evaluate the transmission triggering conditions). Please quantify and comment on these clearly in the paper. 2. Many critical tuning parameters, such as the step-size alpha and the Lyapunov parameters xi are quite complex to tune. Although the authors provide simple parameter selections that satisfy the complex formulas, these parameters are not used in the numerical experiments. Please use the theoretically justified parameters in your experiments (at least as a complement to the present one) and comment on whether the parameters used in the numerical experiments satisfy the theoretical bounds. It would also be nice with some intuition of how you select D. In addition, it would be useful to have some discussion about how sensitive your numerical results are to the different non-trivial parameter choices. 3. It would be nice to discuss the increase in total iteration counts (as a surrogate of wall-clock time) vs the decrease in total communication load. Although this is difficult in general, with your simplified step-size expression, you should also get a simple value for the contraction modulus. You could then compare this one with the one we get for (compressed or uncompressed gradient descent). 4. In Theorem 1, you prove that V decays, but what about f or gradient norm? Can you say something at all about these? 5. Since your quantization scheme is memory-based, I think that you should also compare it against error-compensated gradient quantization schemes (e.g. Alistarh et al., NeurIPS 2018). 6. Only the strongly convex case is analyzed, so the results do not in general apply to Neural Network training. Could you also analyze the non-convex case? If not, I am not sure that it makes much sense to have the simple one-layer ReLU network in the numerical experiments. 7. Some equation references point to supplementary (this is probably a simple LaTeX compilation error, but please revise!) ** Update after rebuttals **. Thank you for your reply. You have addressed most of my concerns and I believe that the numerical experiments consistent with the theory and your refined theoretical analysis will make the final version of the manuscript even better.

[Author Response · NeurIPS 2019]

**Rebuttal for "Communication-Efficient Distributed Learning via Lazily Aggregated Quantized Gradients"**

**Reviewer 1.** My main concern on the theoretical analysis. Provide a more fine-grained analysis. The linear rate constant
can be tightened/simplified, and will be presented it in the final version. With this tighter analysis, the bound explicitly
establishes the dependence of convergence rate on the condition number $\kappa$, namely $\sigma_2 = (1 - \frac{a(\xi)+b(\tau)}{\kappa} + \tau^2 c)^{1/\bar{t}}$,
where constants $a(\xi)$ and $b(\tau)$ increase as $\xi$ and $\tau$ decrease, and all $a(\xi), b(\tau), c$ do not depend on $\kappa, L, \mu$. Clearly,
as quantized values become precise enough ($\tau^2 \to 0$), LAQ approaches the convergence rate of LAG [6], namely,
$\sigma_2 = (1 - \frac{a(\xi)}{\kappa})^{1/\bar{t}}$. If there is no quantization or skipping of communication rounds, $\tau^2 \to 0$, $\xi = 0, \bar{t} = 1$, LAQ
converges with $\sigma_2 = (1 - \frac{a(0)}{\kappa})$, same as GD. We hope the reviewer will appreciate the merits of this analysis.

**Reviewer 2.** 1. The relation of Lyapunov function to the simple risk. Analysis does not apply to neural networks.
Linear convergence of the Lyapunov function also implies that $f(\boldsymbol{\theta}^k) - f(\boldsymbol{\theta}^*)$, $\|\nabla f(\boldsymbol{\theta}^k)\|_2^2$, and $\|\boldsymbol{\theta}^k - \boldsymbol{\theta}^*\|_2^2$, all converge
with a linear rate. From the definition of a Lyapunov function, it is clear that $f(\boldsymbol{\theta}^k) - f(\boldsymbol{\theta}^*) \leq \mathbb{V}(\boldsymbol{\theta}^k) = f(\boldsymbol{\theta}^k) - f(\boldsymbol{\theta}^*) +$
$\sum_{d=1}^{D} \sum_{j=d}^{D} \frac{\xi_j}{\alpha} \|\boldsymbol{\theta}^{k+1-d} - \boldsymbol{\theta}^{k-d}\|_2^2 \leq \sigma_2^k \mathbb{V}^0$, meaning the risk error $f(\boldsymbol{\theta}^k) - f(\boldsymbol{\theta}^*)$ converges linearly. The $L$-smoothness
results in $\|\nabla f(\boldsymbol{\theta}^k)\|_2^2 \leq 2L[f(\boldsymbol{\theta}^k - f(\boldsymbol{\theta}^*)] \leq 2L\sigma_2^k \mathbb{V}^0$; hence, the gradient norm $\|\nabla f(\boldsymbol{\theta}^k)\|_2^2$ also converges linearly.
Similarly, the $\mu$-strong convexity implies $\|\boldsymbol{\theta}^k - \boldsymbol{\theta}^*\|_2^2 \leq \frac{2}{\mu}[f(\boldsymbol{\theta}^k - f(\boldsymbol{\theta}^*)] \leq \frac{2}{\mu}\sigma_2^k \mathbb{V}^0 - \|\boldsymbol{\theta}^k - \boldsymbol{\theta}^*\|_2^2$ also converges linearly.
Convergence analysis of nonconvex nonsmooth objectives is important, and is included in our future research agenda.
2. My only concern is that the experiments have been only conducted on MNIST. If accepted, the final version will
report experiments on SUSY, IJCNN1 and COVTYPE, for which the results are also promising.
3. Connection with [2], [3], [4]. Compared with the innovation-agnostic random selection [2], LAQ explicitly
leverages the gradient innovation in both worker selection and gradient quantization, which will result in more effective
communication reduction. LAQ also differs from the dynamic averaging [3] in their design principles, as [3] skips
communication when the local model does not differ too much from the global model, while LAQ skips communication
when the fresh gradient does not differ too much from the stale one. Lazy aggregation and dynamic averaging can be
jointly leveraged to further reduce communication. Developing lock-free LAQ or asynchronous quantized method based
on [4] is interesting. Due to asynchrony, the resultant algorithm may require re-deriving the communication conditions
(based on a counterpart of Lemma 2), which likely will complicate analysis of the new Lyapunov function. Since it
needs careful investigation, we will tackle it in our future work. A discussion with these references will be added.
4. The proof of them 1, ..., Lyapunov-functions ... not intuitive. The design of Lyapunov function $\mathbb{V}(\boldsymbol{\theta})$ is coupled with
the communication rule (7a) that contains a parameter difference term. Intuitively, if no communication is skipped at
the current iteration, LAQ behaves as GD that decreases the objective residual in $\mathbb{V}(\boldsymbol{\theta})$; if certain uploads are skipped,
LAQ's rule (7a) guarantees that the error of using stale gradients is comparable to the parameter difference in $\mathbb{V}(\boldsymbol{\theta})$ to
ensure its descent. Thus, Lyapunov function always decreases. We will add more intuition in the proof. We incorrectly
marked the reproducibility, but will make the code publicly available. Thank you for the favorable recommendation.

**Reviewer 3.** 1. The scheme requires additional memory. The extra memory is low. The server stores the last aggregated
gradient (dimension $p$), and each worker stores the last gradient (dimension $p$) and $D$ model change norms ($D$ scalars).
2. Many critical tuning parameters, such as the step-size $\alpha$ and the Lyapunov parameters $\xi$ are quite complex to tune.
With the smoothness $L = 19$ in the simulation setting, our parameters used in simulations slightly violate (17). During
the rebuttal period, we conducted a simple experiment with $\xi_d = \xi = 1/160, D = 10$ and $\alpha = 0.01 (\rho = 0.01, \rho_2 = 0.5)$
satisfying (17). It turns out that the result is comparable with that presented in our submission: test accuracy 0.9082,
iteration # 2530, communication # 530, and bit # $1.66 \times 10^7$. To assess the sensitivity of parameters, we tested under
variable $\alpha$ (with $D = 10$) and variable $D$ (with $\alpha = 0.02$) values, as summarized below. Here, $\xi = 0.8/D$ and $\epsilon = 10^{-6}$.

| | $\alpha = 0.01$ | $\alpha = 0.015$ | $\alpha = 0.02$ | $\alpha = 0.04$ | $D = 2$ | $D = 5$ | $D = 10$ | $D = 15$ |
|---|---|---|---|---|---|---|---|---|
| **Iter #** | 5219 | 3459 | 2663 | 1410 | 2448 | 2503 | 2663 | 2749 |
| **Comm #** | 825 | 626 | 618 | 1908 | 534 | 512 | 618 | 678 |
| **Bit # ($\times 10^7$)** | 2.59 | 1.96 | 1.94 | 5.99 | 2.67 | 1.61 | 1.94 | 2.13 |
| **Accuracy** | 0.9082 | 0.9082 | 0.9082 | 0.9082 | 0.9082 | 0.9082 | 0.9082 | 0.9082 |

3. The increase in iteration counts vs the decrease in communication load. With the updated finer-grained analysis, we
can obtain the linear rate constant $\sigma_2 = (1 - \frac{a(\xi)+b(\tau)}{\kappa} + \tau^2 c)^{1/\bar{t}}$, where constants $a(\xi)$ and $b(\tau)$ increase as $\xi$ and $\tau$ decrease,
and all $a(\xi), b(\tau), c$ do not depend on $\kappa, L, \mu$. Thus, $\bar{t} \frac{\kappa}{a(\xi)+b(\tau)-\kappa\tau^2 c} \log(1/\epsilon)$ iterations, or $bp\bar{t} \frac{\kappa}{a(\xi)+b(\tau)-\kappa\tau^2 c} \log(1/\epsilon)$ bits,
are needed to reach $\epsilon$-accuracy. The compressed GD by Khirirat ("Distributed learning with compressed gradients") in a
centralized setup requires the same order of iterations $c_1 \frac{(\mu+\bar{L})^2}{4\mu L} \log(1/\epsilon)$, or $(\log_2 p + bp)c_2 c_1 \frac{(\mu+\bar{L})^2}{4\mu L} \log(1/\epsilon)$ bits; in the
distributed setup, this approach yields a near-optimal solution. Uncompressed methods, e.g., GD, entail $\frac{\kappa+1}{2} \log(1/\epsilon)$
iterations (fewer than LAQ), but more bits (usually encode a float using 32 bits while $b$ bits in LAQ).
4. In Them 1, V decays, what about f or gradient norm? Non-convex case? See our reply to Comment 1 of Reviewer 2.
5. Error compensated schemes. The error compensation schemes skip communicating certain *entries* of the gradient,
but communicate with *all workers*. LAQ skips communicating with certain *workers*, but communicates *all (quantized)*
*entries*. The two are not mutually exclusive, and can be used jointly. We will try to empirically compare with [Alistarh
etal'18] in the final version. Thanks for recognizing the novelty of our work.

[Meta-Review · NeurIPS 2019]

The paper adds quantization to lazily aggregated quantized gradient methods. The contributions were uniformly liked by the reviewers, and the feedback round was productive. We hope the authors will incorporate the detailed reviewer comments in the camera ready version, and in particular will discuss relations to a recent handful of papers on error-compensation/feedback. In the latter, quantization quality is known to only be required in expectation, not in every step, so already allows some communication steps to be skipped.